# Scale-invariant attention

**Ben Anson**
School of Mathematics
University of Bristol
ben.anson@bristol.ac.uk

**Xi Wang**
Department of Computer Science
Johns Hopkins University
xidulu@gmail.com

**Laurence Aitchison**
School of Computer Science
University of Bristol
laurence.aitchison@bristol.ac.uk

## Abstract

One persistent challenge in LLM research is the development of attention mechanisms that are able to generalise from training on shorter contexts to inference on longer contexts. We propose two conditions that we expect all effective long-context attention mechanisms to have: scale-invariant total attention, and scale-invariant attention sparsity. Under a Gaussian assumption, we show that a simple position-dependent transformation of the attention logits is sufficient for these conditions to hold. Experimentally we find that the resulting scale-invariant attention scheme gives considerable benefits in terms of validation loss when zero-shot generalising from training on short contexts to validation on longer contexts, and is effective at long-context retrieval.

## 1   Introduction

One key challenge in modern LLMs is scaling up context length at inference time, while maintaining model performance. We approach this question of length generalisation by considering scale invariance. In particular, we are inspired by the "scale-invariant" statistics of natural images (Van der Schaaf & van Hateren, 1996). Scale invariance, for images, is actually highly intuitive, and means that there is structure at all spatial scales. For example, in an image there might be big features that are 100–1000 pixels across, and some small features that are only 1–10 pixels across. In natural images, features at both spatial scales are important: you cannot remove features at either scale without radically altering the image (Van der Schaaf & van Hateren, 1996).

For attention over text, instead of pixels, we considered *token ranges* at different scales:

- 1–10 tokens in the past (e.g. those in the same sentence),
- 10–100 tokens in the past (e.g. those in the same paragraph),
- 100–1,000 tokens in the past (e.g. those in the same section),
- 1,000–10,000 tokens in the past (e.g. those in the same document), and so on.

With this in mind, we define scale-invariant attention as any attention scheme that satisfies two desiderata: scale-invariant total attention and scale-invariant attention sparsity.

**Scale-invariant total attention** is the property that the sum of attention weights in each of the above ranges is roughly similar. Intuitively, that means that the model attends to both the local context (e.g. 10–100 tokens ago) while at the same time taking account of information from the global context (e.g. 1,000–10,000 tokens ago). Scale-invariant total attention addresses a key issue with attention mechanisms: as the context gets longer, models tend to pay more attention to distant tokens at the

39th Conference on Neural Information Processing Systems (NeurIPS 2025).

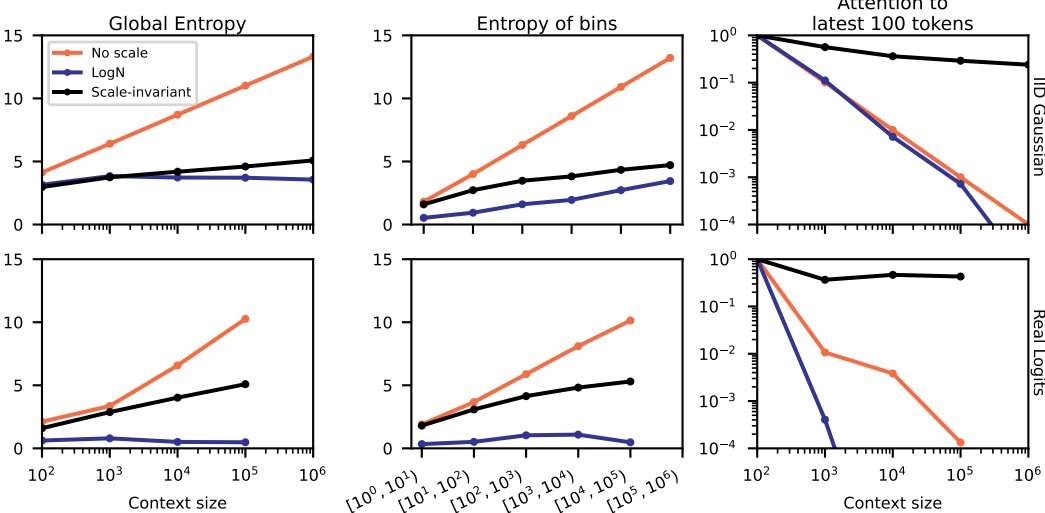

Figure 1: **Scale-invariant attention controls the entropy without sacrificing attention over the local context.** We consider three metrics for attention schemes: (*left*) the global attention entropy, (*middle*) entropy within particular ranges of tokens (e.g. 10–100), and (*right*) total attention to the previous 100 tokens. The top row uses IID Gaussian logits, following our theoretical approach in Sec. 3.2. For LogN, the IID logits are multiplied by $s \log N$, where $N$ is the sequence length and $s = 0.4$. The bottom row uses attention logits sampled from models trained with $p$-RoPE and 'No scale', LogN, and our scale-invariant transform. With no logit scaling, the attention becomes increasingly diffuse as the context grows (i.e. the distribution over logits has high entropy). LogN scaling reduces the entropy and thus ensures that attention remains sparse even at longer contexts. However, LogN still forfeits the ability to attend to the local context (e.g. 100 most recent) tokens. Scale-invariant attention strikes a balance between low entropy and attending over the local context.

expense of the local context (e.g. Fig. 1, right column; the blue and orange lines decay quickly to zero). While scale-invariant total attention doesn't entirely eliminate that issue, it does ensure that the attention paid to the local context shrinks only very slowly as the context length grows (e.g. Fig. 1, right column; the black lines decay to zero much more slowly).

Total attention tells us the amount of attention a certain region receives. However, scale invariance of the total attention does not tell us about how the attention will be distributed among tokens in the region, i.e. whether the attention is spread out among many tokens or concentrated onto only a few tokens. As the region gets wider (e.g. from 10–100 tokens ago to 10,000–100,000 tokens ago), we might expect attention to spread out over more tokens simply due to the increased number of tokens. This "spreading out" of attention is possibly suboptimal for large contexts (Nakanishi, 2025), and that instead, we should focus attention on only a few of the most relevant tokens. To measure these effects, we use the entropy, which roughly captures the logarithm of the number of tokens attended to.

**Scale-invariant attention sparsity** captures this notion. In particular, we define two kinds of scale-invariant attention sparsity. **Strong** scale-invariant attention sparsity implies that the number of tokens attended to in each region is constant. For example, if the model attends to 8 tokens in the 10–100 token range, strong scale-invariant attention sparsity says it will also attend to approximately 8 tokens in the 1,000–10,000 token range. Strong scale-invariant attention sparsity implies an extreme increase in sparsity as the context gets longer that may be difficult to achieve with practical attention mechanisms. We therefore also introduce **weak** scale-invariant attention sparsity, which simply states that the sparsity increases as the context gets longer (i.e. attention is relatively dense in the region from 10–100 tokens, and much sparser in the region from 1,000–10,000, but you still attend to more tokens in the 1,000–10,000 region than the 10–100 region).

Our main contributions are:

- We introduce the concepts of scale-invariant total attention, and weak and strong scale-invariant attention sparsity as desirable properties when attending over long contexts.

- We derive a simple, position-dependent transformation of attention logits that provably satisfies scale-invariant total attention, and empirically satisfies weak scale-invariant attention sparsity for Gaussian logits.
- We implement scale-invariant attention in conjunction with $p$-RoPE (Barbero et al., 2024b). We show that our method, 'scale-invariant $p$-RoPE', exhibits improvements in validation loss both when doing long-context training, and when zero-shot generalising to longer contexts. Our method also matches the performance of the best alternatives in an out-of-distribution long context 'needle-in-a-haystack' task.

## 2   Related work

Handling long sequences effectively in Transformer-based models remains a significant challenge and is of high interest to the deep learning community (Bai et al., 2024; Ye et al., 2024; Jin et al., 2024; Beltagy et al., 2020; Ding et al., 2023; Munkhdalai et al., 2024; Bulatov et al., 2024; Liu et al., 2023a; Barbero et al., 2024a; Hu et al., 2024). Below, we discuss several strategies in this literature relating to our work.

**Long contexts via the positional encoding**: Methods like ALiBi (Press et al., 2021) introduce a static, causal bias directly into the attention logits. ALiBi endows the model with an inductive bias towards recent tokens (Kazemnejad et al., 2023), thus helping with longer contexts. Rotary Position Embeddings (RoPE) (Su et al., 2024) have emerged as the most popular position encoding scheme, encoding relative positions. RoPE does not generalise to longer sequences out of the box (Peng et al., 2023), leading to subsequent research improving RoPE's length extrapolation capabilities (Zhu et al., 2023; Wang et al., 2024). Many achieve this by modifying RoPE's frequency spectrum; for example, Positional Interpolation (PI) (Chen et al., 2023), NTK-aware scaling (bloc97, 2023), YaRN (Peng et al., 2023), and by simply increasing the RoPE base $\theta$ parameter (Grattafiori et al., 2024; Gemma Team, 2025; Barbero et al., 2024b; Liu et al., 2023b).

**Entropy Control**: Beyond positional information, the properties of the attention distribution itself are crucial, especially in long contexts where attention scores can "smear"/"spread out" across many tokens. Scalable-Softmax (SSMax), also known as the 'LogN trick' (Nakanishi, 2025; Chiang & Cholak, 2022; Jianlin, 2021; Bai et al., 2023; Llama 4 Team, 2025), addresses this by multiplying the attention logits by $s \log N$, where $N$ is the context length and $s$ is a learned scale parameter. This multiplier has the effect of sharpening/focusing the attention distribution. Li et al. (2025) adopt a similar approach: controlling the entropy of the attention distribution for better length generalization.

These prior works are perhaps the most similar to our approach. However, the key issue with such approaches is that they treat the local context (e.g. previous 10–100 tokens) in the same way as the global context (e.g. 10,000 tokens ago). As the context gets larger, the attention paid to the 100 most recent tokens drops rapidly for the prior approaches, but stays markedly more consistent with scale-invariant attention (e.g. for LogN see Fig. 1). In contrast, we started off by carefully specifying how we wanted attention to behave in the local and global contexts (Sec. 3.1) by giving the scale-invariant total attention and scale-invariant attention sparsity desiderata. This means that e.g. LogN has a position-independent multiplicative bias, whereas our approach has position-dependent multiplicative and additive biases for the logits.

**Efficient Long-Context Training and Inference**: A key observation of Xiong et al. (2023) is that continual pretraining on long contexts, after initial pretraining on shorter sequences, is often sufficient and much more computationally efficient. Thus, a simple strategy for improving long-context performance, given sufficient computational budget, is continual pretraining on longer contexts. This has been adopted widely (Grattafiori et al., 2024; Gao et al., 2024; Lieber et al., 2024; Yang et al., 2024; Cohere Team, 2025; Llama 4 Team, 2025; Liu et al., 2024a). Of course, this strategy considerably increases the complexity and memory cost of the pretraining pipeline, making approaches like ours that can zero-shot generalise very valuable. Furthermore, one might expect long-context training to be far easier (e.g. requiring fewer long-context training steps for optimal performance) for approaches that already have good long-context performance due to zero-shot generalisation.

For inference on sequences exceeding the trained context length, alternative strategies bypass applying dense attention over the entire context, allowing for 'infinite attention' (Munkhdalai et al., 2024; Liu et al., 2024b; Martins et al., 2021; Chen et al., 2025; Ding et al., 2023). These include maintaining a fixed-size attention window (Beltagy et al., 2020), retrieving relevant context tokens

using heuristics like top-K proximity (Han et al., 2023), or vector similarity search akin to episodic memory systems (Fountas et al., 2024; Xiao et al., 2024). While effective, these approaches operate at a higher level, managing context rather than modifying the core attention mechanism's ability to process it directly, which is the focus of our work.

## 3 Methods

Following FlexAttention (Dong et al., 2024), we define the attention "score" as the dot product of a query $q$ and keys $K$,

$$S_t = \frac{1}{\sqrt{d}} \sum_{\lambda=1}^{d} q_\lambda K_{t\lambda}. \tag{1}$$

Here, $d$ is the head dimension and $\lambda \in \{1, \ldots, d\}$ indexes the feature, and $t$ indexes the position. We are using an unusual form for the sequence indices, but this simplifies notation later. Specifically, we consider a single, fixed query token $q$. Then, $t \geq 0$ into the keys/values from previous tokens. Critically, $t$ counts backwards from the query token, e.g. $t = 1$ indicates the previous token. Logits are computed by applying an attention modifier function to the score,

$$L_t = L(S_t; t). \tag{2}$$

Here, $L_t$ is the actual value of the attention logits for the token $t$ steps back from the query, while $L(S_t; t)$ is the function used to compute those logits. Thus, $L(S_t; t) = S_t$ recovers standard attention.

For use later, we define the unnormalised attention weights $\tilde{A}_t$, (normalised) attention weights $A_t$, and normalisers $Z$, for a sequence of length $T$, as,

$$\tilde{A}_t = \exp(L_t) \qquad A_t = \frac{\tilde{A}_t}{Z} \qquad Z = \sum_{t=1}^{T} \tilde{A}_t. \tag{3}$$

### 3.1 Formal definitions of scale-invariant total attention and attention sparsity

**Scale-invariant total attention.** In our examples we have considered ranges of tokens, 10–100, 100–1,000, 1,000–10,000, 10,000–100,000, etc.. Scale-invariant total attention is the property that the total attention in each of these ranges is somewhat similar, such that the attention allotted to any one range does not dominate the others. We give a formal definition in Def. 3.1, where using $\Delta = 10$ gives the ranges from the examples.

**Definition 3.1** (Scale-invariant total attention). *Consider a set of random variables, $\{L_t\}$, representing attention logits. The total unnormalised attention in range $t_1$ to $t_2$ is,*

$$Z_{t_1}^{t_2} = \sum_{t=t_1}^{t_2-1} \tilde{A}_t = \sum_{t=t_1}^{t_2-1} \exp(L_t). \tag{4}$$

*We say that the total attention is scale-invariant if, for any integer $\Delta > 1$, the expected total unnormalised attention in the range $\{t, t+1, \ldots, t\Delta - 1\}$ is asymptotically constant. That is,*

$$\mathrm{E}\left[Z_t^{t\Delta}\right] = \Theta(1) \quad as \quad t \to \infty. \tag{5}$$

The "$\Theta(1)$" is big-$\Theta$ notation, and means there exist constants $c_1, c_2 > 0$ and $t_0$ such that for all $t > t_0$, $c_1 \leq \mathrm{E}\left[Z_t^{t\Delta}\right] \leq c_2$.

**Scale-invariant unnormalised attention sparsity.** Remember that scale-invariant attention sparsity means that attention should be denser for the local context (e.g. 10–100 tokens ago) and sparser for the global context (e.g. 1,000–10,000) tokens ago. To measure attention sparsity, we consider the number of tokens attended to in a region. To evaluate the number of tokens attended to, one approach is to use the entropy over tokens. We measure the sparsity in the region from $t_1$ to $t_2$, using the entropy for the distribution over tokens in this region,

$$H_{t_1}^{t_2} = -\sum_{t=t_1}^{t_2-1} \frac{\tilde{A}_t}{Z_{t_1}^{t_2}} \log\left(\frac{\tilde{A}_t}{Z_{t_1}^{t_2}}\right) = -\frac{\sum_{t=t_1}^{t_2-1} \tilde{A}_t \log \tilde{A}_t}{Z_{t_1}^{t_2}} + \frac{\log Z_{t_1}^{t_2} \sum_{t=t_1}^{t_2-1} \tilde{A}_t}{Z_{t_1}^{t_2}}, \tag{6}$$

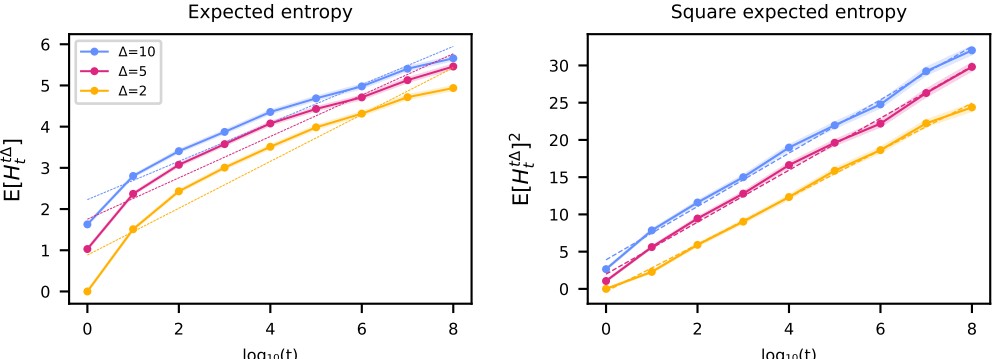

Figure 2: **Expected entropy of scale-invariant attention at different scales is sub-logarithmic**. Here, we sample sequences of independent standard Gaussian logits, and apply the scale-invariant attention transformation. We estimate the expected entropy in ranges $[t, t\Delta)$, where the size of the range is controlled by $t$ (x-axis) and $\Delta$ (line color). We see that this expected entropy measure scales sub-logarithmically (left), and with the right plot suggesting a $\sim \sqrt{\log(t)}$ scaling. The dashed lines show a best linear fit.

where $A_t$ is defined in Eq. (3). Defining the unnormalised negentropy as,

$$\tilde{N}_{t_1}^{t_2} = \sum_{t=t_1}^{t_2-1} \tilde{A}_t \log \tilde{A}_t, \tag{7}$$

and remembering the definition $Z_{t_1}^{t_2}$ (Eq. 4), we can then write the entropy in range $t_1$ to $t_2$ as,

$$H_{t_1}^{t_2} = -\frac{\tilde{N}_{t_1}^{t_2}}{Z_{t_1}^{t_2}} - \log Z_{t_1}^{t_2}. \tag{8}$$

Thus, it seems reasonable to consider the behavior of the unnormalised negentropy (Eq. 7), because if $\tilde{N}_t^{t\Delta}$ and $Z_t^{t\Delta}$ are asymptotically constant as $t \to \infty$, then by Eq. (8) we expect $H_{t_1}^{t_2}$ to also be asymptotically constant. Thus we define:

**Definition 3.2** (Scale-invariant unnormalised attention sparsity). *Consider a set of random variables, $\{L_t\}$, representing attention logits. We say that we have scale-invariant unnormalised attention sparsity if, for any integer $\Delta > 0$,*

$$\mathrm{E}\left[\tilde{N}_t^{t\Delta}\right] = \Theta(1) \quad as \quad t \to \infty. \tag{9}$$

*where $\tilde{N}_t^{t\Delta}$ is the unnormalised negentropy (Eq. 7).*

**Weak and strong scale-invariant attention sparsity.** While the argument above suggests that scale-invariant unnormalised attention sparsity is an important property, ultimately we are interested in giving formal definitions of weak and strong attention sparsity.

We define weak scale-invariant attention sparsity such that as input lengths increase — for example, from 10–100 to 100–1,000 tokens — the number of attended tokens grows sublinearly. In contrast, standard attention with unscaled logits yields linear growth. Since entropy is roughly the log of the number of attended tokens (a uniform distribution over $k$ tokens has entropy equal to $\log k$), weak sparsity requires sublinear growth in entropy with respect to $\log(t)$ (see Definition 3.3).

**Definition 3.3** (Weak scale-invariant attention sparsity). *Consider a set of random variables, $\{L_t\}$, representing attention logits. We say that the attention sparsity is weakly scale-invariant if, for any integer $\Delta > 1$,*

$$\mathrm{E}\left[H_t^{t\Delta}\right] = o(\log t) \quad as \quad t \to \infty. \tag{10}$$

Remember $o(\log(t))$ is 'little-$o$' notation which means that $\mathrm{E}\left[H_t^{t\Delta}\right]$ scales strictly slower than $\log(t)$. Strong scale-invariant attention sparsity implies that the number of tokens attended to is asymptotically constant as we go from e.g. the past 10–100 tokens to the past 1,000–10,000.

**Definition 3.4** (Strong scale-invariant attention sparsity). *Consider a set of random variables, $\{L_t\}$, representing attention logits. We say that the attention sparsity is strongly scale-invariant if, for any integer $\Delta > 1$,*

$$\mathrm{E}\left[H_t^{t\Delta}\right] = \Theta(1) \quad as \quad t \to \infty, \tag{11}$$

*i.e. we expect $H_t^{t\Delta}$ to be asymptotically constant as $t \to \infty$.*

## 3.2 What characteristics of the logits are required for scale-invariant attention?

Next, we ask what properties would be sufficient for scale-invariant total attention and for some form of scale-invariant attention sparsity. Lemma 1 gives scale-invariant total attention, and Lemma 2 gives scale-invariant unnormalised attention (see Appendix D for the proofs).

**Lemma 1.** *Consider a set of random variables, $\{L_t\}$, representing attention logits. Let $\tau > 0$ be a lengthscale parameter, and $\alpha > 0$ a multiplicative constant. If the attention logits satisfy,*

$$\mathrm{E}\left[\tilde{A}_t\right] = \frac{\alpha}{t/\tau + 1}, \tag{12}$$

*then we have scale-invariant total attention (Def. 3.1).*

**Lemma 2.** *Consider a set of random variables, $\{L_t\}$, representing attention logits. Let $\tau > 0$ be a lengthscale parameter, and $\beta > 0$ a multiplicative constant. If the attention logits satisfy,*

$$\mathrm{E}\left[\tilde{A}_t \log \tilde{A}_t\right] = \frac{\beta}{t/\tau + 1}, \tag{13}$$

*then we have scale-invariant unnormalised attention sparsity (Def. 3.2).*

To construct an attention mechanism that satisfies Eq. (12) and Eq. (13), we consider a simplified setting in which the logits are marginally Gaussian and arise from taking Gaussian "base logits", $\bar{L}_t \sim \mathcal{N}(0, 1)$, and transforming them by multiplying by $a_t$ and adding a bias, $m_t$,

$$L_t = a_t \bar{L}_t + m_t \sim \mathcal{N}(m_t, a_t^2). \tag{14}$$

Our goal is to find $m_t$ and $a_t^2$ such that scale-invariant total attention and scale-invariant unnormalised attention sparsity hold. In particular that requires,

$$\frac{\alpha}{\frac{t}{\tau} + 1} = \mathrm{E}\left[\tilde{A}_t\right] = e^{m_t + a_t^2/2}, \tag{15a}$$

$$\frac{\beta}{\frac{t}{\tau} + 1} = \mathrm{E}\left[\tilde{A}_t \log \tilde{A}_t\right] = (m_t + a_t^2)e^{m_t + a_t^2/2}, \tag{15b}$$

where $\tilde{A}_t = \exp(L_t)$. Solving for $m_t$ and $a_t$, we have (see Appendix C for details),

$$a_t = \sqrt{2\left[\log(t/\tau + 1) - \log \alpha + \beta/\alpha\right]}, \tag{16a}$$

$$m_t = -a_t^2 + \beta/\alpha. \tag{16b}$$

For this solution to be valid, we only require $\beta \geq \alpha \log \alpha$ since $\log(t/\tau + 1) \geq 0$ when $t \geq 0$. We formally summarise the above results in Theorem 1 (proof in Appendix F), which tells us that this approach does indeed give scale-invariant total attention and scale-invariant unnormalised attention sparsity.

**Theorem 1.** *Suppose attention logits $\{L_t\}$ are marginally Gaussian with mean $m_t$ and standard deviation $a_t$ defined by Eq. (16). Assuming $\alpha, \beta, \tau > 0$, $\beta \geq \alpha \log \alpha$, then we have scale-invariant total attention and scale-invariant unnormalised attention sparsity.*

We therefore propose to scale the logits in real attention using $a_t$ and $m_t$ defined in Eq. (16). As $t$ increases, the variance, $a_t^2$, *increases* as the logarithm of $t$, while the mean *decreases* as the logarithm of $t$.

Finally, note that while we have proven that we have scale-invariant unnormalised attention with this choice of $a_t$ and $m_t$, we have not proven that we have strong or weak scale-invariant attention. We therefore checked empirically whether using IID Gaussian logits, scaled by $a_t$ and $m_t$, gave weak or

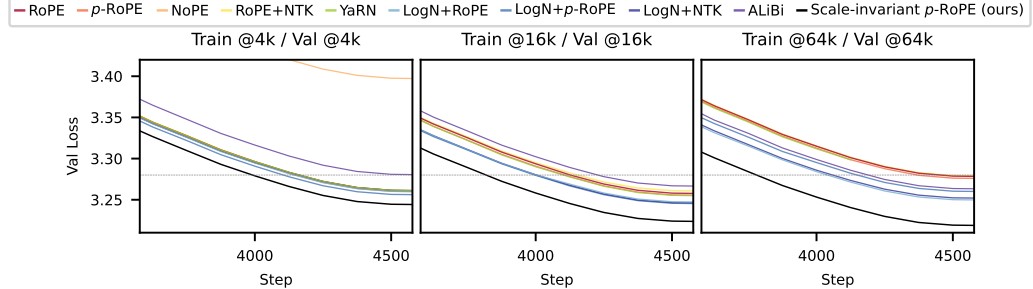

(a) Scale-invariant attention improves language modelling over a range of training context lengths.

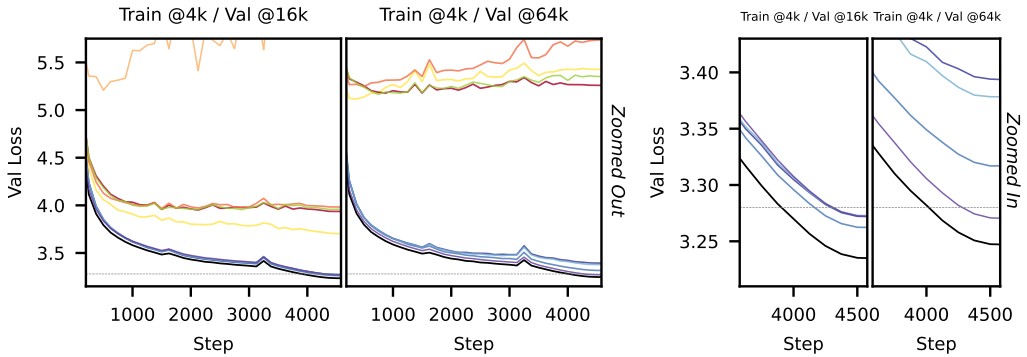

(b) Scale-invariant attention enables zero-shot long-context generalization of at least 16x.

Figure 3: Validation losses throughout training of a 162M parameter GPT-2-like model with different attention mechanisms (our scale-invariant scheme shown in black). NoPE is omitted in all but top-left and bottom-left panels to avoid excessive zooming out (due to high loss). The gray dashed line shows the baseline of 3.28. The validation loss for many methods increases in unison at steps ∼1500 and ∼3250 (see (b), left) despite aggregating over seeds; this is due to a fixed training and validation data ordering.

strong scale-invariant attention entropy (Fig. 2). We find that $H_t^{t\Delta}$ appears to scale with $\sqrt{\log(t)}$, and hence seems to satisfy weak, but not strong scale-invariant attention sparsity.

**Hyperparameters.** Introducing $a_t$ and $m_t$ (Eq. 16), appears to have introduced three additional hyperparameters to tune. We reduce this to one additional hyperparameter, the lengthscale $\tau$, by specifying a boundary condition — $a_0^2 = 1$ and $m_0 = 0$. This boundary condition corresponds to not changing the scale of the local tokens. Substituting into Eq. (16b) gives $0 = -1 + \beta/\alpha$. Similarly Eq. (16a) requires that $1 = -2\log\alpha + 2\beta/\alpha$. Putting these together, we obtain $\alpha = \beta = e^{0.5}$.

That leaves us with the lengthscale as the only hyperparameter. Note that when $t$ is small relative to $\tau$, neither $a_t^2$ nor $m_t$ change much. Therefore, the timescale sets the size of a local region in which attention is approximately unscaled. Intuitively, it therefore makes sense to choose $\tau$ somewhere in the region of 1–100 tokens. In Appendix H we tried $\tau \in \{10^{-2}, 10^{-1}, 10^0, 10^1, 10^2\}$ and find that $\tau = 10$ performs best in practice.

## 4 Experiments

In this section, we compare scale-invariant attention with other dense attention methods including Dynamic NTK interpolation (RoPE+NTK) (bloc97, 2023), LogN scaling/SSMax (Nakanishi, 2025), $p$-RoPE (Barbero et al., 2024b), and ALiBi (Press et al., 2021). Our results show that our method, scale-invariant $p$-RoPE, has uniformly lower validation loss at a variety of training lengths (4k, 16k, 64k). Additionally, scale-invariant $p$-RoPE demonstrates stronger zero-shot long-context generalization (e.g. in 'Train @4k/Val @16k' and 'Train @4k/Val @64k' settings) versus all other methods. Finally, scale-invariant $p$-RoPE, along with LogN, saturated "needle-in-a-haystack" task.

We pretrained GPT-2-style models (Radford et al., 2019) (with QK-norm, ReLU$^2$ activations, etc. (Jordan et al., 2024a)) from scratch on the FineWeb dataset (Penedo et al., 2024), using a fixed training data ordering and a 10M token validation set. We trained linear layers with Muon (Jordan et al., 2024b), and remaining parameters with Adam. We implemented scale-invariant attention using FlexAttention (Dong et al., 2024).

We trained two models: one with 162M parameters and another with 304M. The 162M one were trained for 4578 steps on 2.4B tokens over a range of context lengths (4k, 16k, 64k). The 304M parameter models were trained only on the best length-generalising methods, for 10.9k steps on 10B tokens, at 4k context length. We give further experimental details in Appendix G. For the 162M parameter model, we targeted a validation loss of 3.28, following Karpathy's GPT-2 reproduction (Karpathy, 2024; Jordan et al., 2024a), shown in the figures as a horizontal grey dashed line.

**Long-context performance.** We examined in-distribution, long-context performance of the different attention methods by looking at the validation loss for the same context length used for training. Even in this in-distribution setting, scale-invariant $p$-RoPE shows strong improvements in validation loss at all training lengths, 4k, 16k, and 64k (Fig. 3a).

**Length Generalization.** We evaluate length generalization by measuring validation loss on long sequences (16k and 64k) when training on 4k tokens. The 'Train @4k/Val @64k' setting in particular represents a considerable jump of 16× between train and validation. Table 1 and Fig. 3b report results for the 162M model. In the left panel of the Figure, ALiBi, LogN, and our method substantially outperform other approaches in generalizing to longer sequences. The right panel zooms in and shows that scale-invariant $p$-RoPE achieves the strongest generalization overall.

In preliminary experiments we tried other scale-invariant methods. We found that the most obvious method, scale-invariant RoPE, did not generalise well to long contexts (see Appendix I.3). The $p$-RoPE method is similar to RoPE but excludes low-frequency/high-wavelength components in the position embedding, possibly suggesting that low-frequency components in RoPE interfere with the scale-invariant transformation $L_t \mapsto a_t L_t + m_t$. LogN also demonstrates stronger performance when paired with $p$-RoPE rather than RoPE. We were surprised that RoPE+NTK struggled to generalise in the 'Train @4k / Val @64k' setting, but we believe this can be explained by the training context size: 'Train @16k / Val @64k' is much better for RoPE+NTK (see Fig. 7 in the Appendix).

Fig. 4 presents pretraining losses for ALiBi, LogN+$p$-RoPE, and our method on a larger 304M model — selected due to their performance on the 162M 'Train @4k/Val @64k' task. Scale-invariant $p$-RoPE maintains its advantage at this larger scale.

**Needle in a Haystack.** The key benefit of scale-invariant attention is that it balances local and sparse global attention. As such, we might worry that long-context retrieval performance might suffer versus other approaches that do not have specific mechanisms to ensure that attention to the local context does not vanish.

To assess whether long-context information retrieval capabilities suffered, we fine-tuned models on a needle-in-a-haystack task (note that fine-tuning on this task is unusual, but we found that prompting alone was not sufficient to perform this task, as bigger models/more pretraining would be required). Needle-in-a-haystack (Kamradt, 2023) measures a model's ability to precisely retrieve specific details (needles) from a large body of text (the haystack). Our tasks constructs prompts by concatenating text samples from the C4 dataset (Roberts et al., 2019) and embedding "needles" of the form 'The special magic <city> number is <7_digit_number>'. We insert three (rather than one) needles uniformly, at random, into each context for more signal per example, with each needle contributing separately to the overall accuracy. A successful retrieval requires the model to output both the city and the associated number correctly. We trained models on sequences of length 4k, and tested at 4k, 16k, and 64k.

Table 2 show that scale-invariant $p$-RoPE and LogN+$p$-RoPE perform well, while almost all other methods fail almost completely at 64k context length. Thus, despite focusing more on local context, our method does not seem to have suffered in retrieval performance.

Table 1: Final mean validation losses ($\pm 1$ standard error across 3 seeds) for different methods when training with 4k context length on a 162M parameter GPT-2-style model. The error bars are small due to consistent training data ordering, and a fixed validation set.

| Method | Val @ 4k | Val @ 16k | Val @ 64k |
|---|---|---|---|
| RoPE | $3.261 \pm 0.001$ | $3.936 \pm 0.010$ | $5.260 \pm 0.014$ |
| $p$-RoPE | $3.260 \pm 0.001$ | $3.984 \pm 0.008$ | $5.735 \pm 0.085$ |
| NoPE | $3.397 \pm 0.000$ | $6.430 \pm 0.059$ | $8.125 \pm 0.062$ |
| RoPE+NTK | $3.261 \pm 0.001$ | $3.703 \pm 0.007$ | $5.430 \pm 0.026$ |
| YaRN | $3.261 \pm 0.000$ | $3.958 \pm 0.018$ | $5.353 \pm 0.065$ |
| LogN+RoPE | $3.260 \pm 0.001$ | $3.273 \pm 0.004$ | $3.378 \pm 0.011$ |
| LogN+$p$-RoPE | $3.256 \pm 0.001$ | $3.262 \pm 0.002$ | $3.317 \pm 0.005$ |
| LogN+NTK | $3.261 \pm 0.001$ | $3.272 \pm 0.002$ | $3.394 \pm 0.025$ |
| ALiBi | $3.281 \pm 0.001$ | $3.272 \pm 0.001$ | $3.270 \pm 0.000$ |
| Scale-invariant $p$-RoPE (ours) | $\mathbf{3.244} \pm 0.001$ | $\mathbf{3.235} \pm 0.001$ | $\mathbf{3.247} \pm 0.001$ |

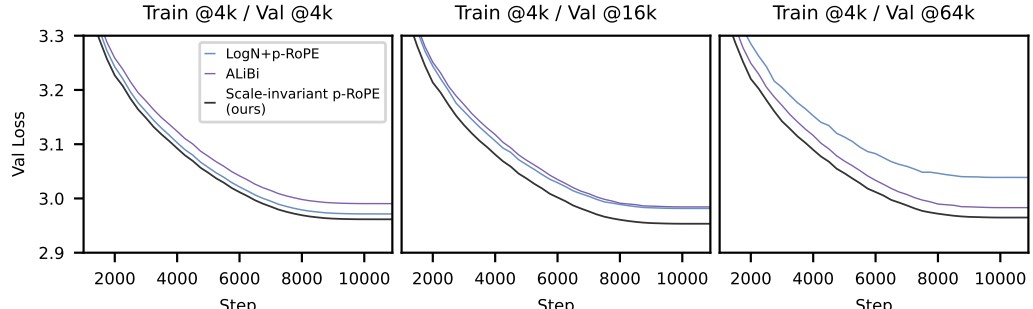

Figure 4: Validation losses throughout training for a 304M parameter model.

Table 2: Mean validation accuracies on the needle-in-a-haystack task, $\pm 1$ standard error. Metrics were calculated over 3 seeds, after 300 steps of fine-tuning.

| Method | Val Acc @4k | Val Acc @16k | Val Acc @64k |
|---|---|---|---|
| RoPE | $0.962 \pm 0.003$ | $0.000 \pm 0.000$ | $0.000 \pm 0.000$ |
| $p$-RoPE | $0.966 \pm 0.001$ | $0.250 \pm 0.020$ | $0.000 \pm 0.000$ |
| NoPE | $0.964 \pm 0.000$ | $0.303 \pm 0.239$ | $0.000 \pm 0.000$ |
| RoPE+NTK | $0.962 \pm 0.001$ | $0.217 \pm 0.078$ | $0.000 \pm 0.000$ |
| YaRN | $\mathbf{0.969} \pm 0.001$ | $0.000 \pm 0.000$ | $0.000 \pm 0.000$ |
| LogN+RoPE | $0.965 \pm 0.002$ | $0.276 \pm 0.010$ | $0.064 \pm 0.006$ |
| LogN+$p$-RoPE | $\mathbf{0.969} \pm 0.002$ | $0.962 \pm 0.003$ | $0.939 \pm 0.009$ |
| LogN+NTK | $0.962 \pm 0.001$ | $0.253 \pm 0.015$ | $0.056 \pm 0.008$ |
| ALiBi | $0.957 \pm 0.002$ | $0.020 \pm 0.001$ | $0.003 \pm 0.000$ |
| Scale-invariant $p$-RoPE (ours) | $0.965 \pm 0.000$ | $\mathbf{0.969} \pm 0.004$ | $\mathbf{0.969} \pm 0.005$ |

## 5 Limitations

In this work, we evaluated our methods by pretraining with 162M and 304M parameter models, and investigated 7B parameter models via continual pretraining in Appendix I.5. Ideally, we would have pretrained from scratch at the multi-billion parameter scale, in line with contemporary state-of-the-art LLMs. While compute resource constraints prevented this, our results, together with the natural theoretical approach, offer no indication that our conclusions would fail to generalise to larger, commercially deployed models.

We focused on scale-invariant $p$-RoPE with dense attention, but the extension to other settings is a promising direction for future investigation.

## 6 Conclusions

We have proposed two desirable properties of attention mechanisms, "scale-invariant total attention" and "scale-invariant attention sparsity", and presented a straightforward modification to attention logits that enables these properties in practice. In our experiments, we found that our scale-invariant attention modification, especially when paired with $p$-RoPE, substantially improves long-context language modelling performance and gives zero-shot generalisation from training on short contexts to testing on long contexts.

## 7 Acknowledgements

We sincerely thank the Engineering and Physical Sciences Research Council and the COMPASS CDT for funding Ben Anson at the University of Bristol. Many of our experiments used computational resources at the Advanced Computing Research Centre at the University of Bristol, with GPUs generously funded by Dr. Stewart.

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

## A    Relating the Indicies in the Paper to Standard Indices

The standard form for attention is,

$$S^{ij} = \frac{1}{\sqrt{d}} \sum_{\lambda=1}^{d} Q_{i\lambda} K_{j\lambda}, \tag{17}$$

where $\lambda$ indexes the feature, $i$ indexes the query (i.e. the token we're generating now) and $j$ indexes the key (i.e. the token we're attending to). We fix $i$, and take $t = i - j$, so

$$S^{ij} = S_{t=i-j} \tag{18}$$

where $S_{t=i-j}$ is given in Eq. 1 in the main text. Then,

$$L^{ij} = L(S^{ij}; i - j) = L(S_t; t) = L_{t=i-j}, \tag{19}$$

where $L_{t=i-j}$ is given in Eq. 1 in the main text. Then the unnormalised attention weights, $\tilde{A}^{ij}$, normalised attention weights, $A^{ij}$ and normalisers, $Z^i$ are,

$$\tilde{A}^{ij} = \exp\left(L^{ij}\right), \tag{20}$$

where, $\tilde{A}_{t=i-j}$ is given in Eq. 3 in the main text.

$$A^{ij} = \frac{\tilde{A}^{ij}}{Z^i}, \tag{21}$$

where, $A_{t=i-j}$ is given in Eq. 3 in the main text.

$$Z^i = \sum_{j=1}^{i} \tilde{A}^{ij} = Z, \tag{22}$$

where, $Z$ is given in Eq. 3 in the main text.

## B    $\mathrm{E}\left[X^k \exp(\alpha X)\right]$ where $X$ is Gaussian

Assume $X \sim \mathcal{N}(\mu, \sigma^2)$, then we can compute the expectation in terms of a moment of a Gaussian,

$$\mathbb{E}[X^k e^{\alpha X}] = \int_{-\infty}^{\infty} x^k e^{\alpha x} \frac{1}{\sqrt{2\pi\sigma^2}} e^{-\frac{(x-\mu)^2}{2\sigma^2}} dx \tag{23}$$

$$= \int_{-\infty}^{\infty} x^k \frac{1}{\sqrt{2\pi\sigma^2}} e^{\alpha x - \frac{(x-\mu)^2}{2\sigma^2}} dx \tag{24}$$

$$= \int_{-\infty}^{\infty} x^k \frac{1}{\sqrt{2\pi\sigma^2}} e^{-\frac{1}{2\sigma^2}(x^2 - 2\mu x + \mu^2 - 2\alpha\sigma^2 x)} dx \tag{25}$$

$$= \int_{-\infty}^{\infty} x^k \frac{1}{\sqrt{2\pi\sigma^2}} e^{-\frac{1}{2\sigma^2}(x^2 - 2(\mu+\alpha\sigma^2)x + \mu^2)} dx \tag{26}$$

$$= e^{\frac{\alpha^2\sigma^2}{2} + \alpha\mu} \int_{-\infty}^{\infty} x^k \frac{1}{\sqrt{2\pi\sigma^2}} e^{-\frac{(x-(\mu+\alpha\sigma^2))^2}{2\sigma^2}} dx \tag{27}$$

$$= e^{\frac{\alpha^2\sigma^2}{2} + \alpha\mu} \mathrm{E}\left[\tilde{X}^k\right], \tag{28}$$

where $\tilde{X} \sim \mathcal{N}(\mu', \sigma'^2) = \mathcal{N}(\mu + \alpha\sigma^2, \sigma^2)$. In the case $\alpha = k = 1$, we have,

$$\mathrm{E}\left[X \exp(X)\right] = (\mu + \sigma^2)\exp(\mu + \sigma^2/2). \tag{29}$$

In the case $\alpha = k = 2$, we have,

$$\mathrm{E}\left[X^2 \exp(2X)\right] = ((\mu + 2\sigma^2)^2 + \sigma^2)\exp(2\mu + 2\sigma^2). \tag{30}$$

# C  Deriving $a_t$ and $m_t$ in the Logit Transformation

From Eq. (15), we wish to find $a_t$ and $m_t$ such that,

$$\frac{\alpha}{\frac{t}{\tau}+1} = e^{m_t + a_t^2/2} \tag{31}$$

$$\frac{\beta}{\frac{t}{\tau}+1} = (m_t + a_t^2)e^{m_t + a_t^2/2}. \tag{32}$$

We begin by dividing Eq. (32) by Eq. (31),

$$\frac{\beta}{\alpha} = m_t + a_t^2, \tag{33}$$

which can be rearranged to,

$$m_t = \frac{\beta}{\alpha} - a_t^2, \tag{34}$$

$$\text{and } a_t^2 = \frac{\beta}{\alpha} - m_t. \tag{35}$$

Now, taking the log of Eq. (31), we have,

$$m_t + \tfrac{1}{2}a_t^2 = -\log\left(\tfrac{t}{\tau}+1\right) + \log\alpha. \tag{36}$$

Define $f_t = \log\left(\tfrac{t}{\tau}+1\right) - \log\alpha$. Then to solve for $a_t^2$, we substitute $m_t$ from Eq. (34) into Eq. (36),

$$\left(\tfrac{\beta}{\alpha} - a_t^2\right) + \tfrac{1}{2}a_t^2 = -f_t \tag{37}$$

$$\tfrac{\beta}{\alpha} - \tfrac{1}{2}a_t^2 = -f_t \tag{38}$$

$$a_t^2 = 2\left(f_t + \tfrac{\beta}{\alpha}\right). \tag{39}$$

To solve for $m_t$, we substitute $a_t^2$ from Eq. (35) into Eq. (36)

$$m_t + \tfrac{1}{2}\left(\tfrac{\beta}{\alpha} - m_t\right) = -f_t \tag{40}$$

$$\tfrac{1}{2}m_t + \tfrac{\beta}{2\alpha} = -f_t \tag{41}$$

$$m_t = -2\left(f_t + \tfrac{\beta}{2\alpha}\right), \tag{42}$$

which can also be written as,

$$m_t = -a_t^2 + \tfrac{\beta}{\alpha}. \tag{43}$$

Thus, we have,

$$a_t^2 = 2\left[\log(t/\tau + 1) + \beta/\alpha - \log\alpha\right] \tag{44}$$

$$m_t = -2\left[\log(t/\tau + 1) + \beta/\alpha - \log\alpha\right] + \beta/\alpha \tag{45}$$

$$= -2\log(t/\tau + 1) - \beta/\alpha + 2\log\alpha. \tag{46}$$

# D  Proofs of Lemmas 1 and 2

**Lemma 1.** *Consider a set of random variables, $\{L_t\}$, representing attention logits. Let $\tau > 0$ be a lengthscale parameter, and $\alpha > 0$ a multiplicative constant. If the attention logits satisfy,*

$$\mathrm{E}\left[\tilde{A}_t\right] = \frac{\alpha}{t/\tau + 1}, \tag{12}$$

*then we have scale-invariant total attention (Def. 3.1).*

*Proof.* We want to show that $\mathrm{E}\left[Z_{t_1}^{t_1\Delta}\right] = \Theta(1)$ as $t_1 \to \infty$. By definition and linearity of expectation:

$$\mathrm{E}\left[Z_{t_1}^{t_1\Delta}\right] = \mathrm{E}\left[\sum_{t=t_1}^{t_1\Delta-1} \tilde{A}_t\right] = \sum_{t=t_1}^{t_1\Delta-1} \mathrm{E}\left[\tilde{A}_t\right]. \tag{47}$$

Using the given condition $\mathrm{E}\left[\tilde{A}_t\right] = \frac{\alpha}{t/\tau+1}$:

$$\mathrm{E}\left[Z_{t_1}^{t_1\Delta}\right] = \sum_{t=t_1}^{t_1\Delta-1} \frac{\alpha}{t/\tau+1} = \alpha\tau \sum_{t=t_1}^{t_1\Delta-1} \frac{1}{t+\tau} = \alpha\tau \sum_{k=t_1+\tau}^{t_1\Delta-1+\tau} \frac{1}{k}. \tag{48}$$

Using the standard integral bounds for the harmonic sum derived by comparison with the integral (see Appendix E):

$$\ln\left(\frac{t_1\Delta+\tau}{t_1+\tau}\right) \leq \sum_{k=t_1+\tau}^{t_1\Delta-1+\tau} \frac{1}{k} \leq \ln\left(\frac{t_1\Delta-1+\tau}{t_1+\tau-1}\right). \tag{49}$$

As $t_1 \to \infty$:

$$\frac{t_1\Delta+\tau}{t_1+\tau} \to \frac{t_1\Delta}{t_1} = \Delta \tag{50}$$

$$\frac{t_1\Delta-1+\tau}{t_1+\tau-1} \to \frac{t_1\Delta}{t_1} = \Delta \tag{51}$$

So, the logarithm terms in both the lower and upper bounds approach $\ln(\Delta)$. By the Squeeze Theorem:

$$\sum_{k=t_1+\tau}^{t_1\Delta-1+\tau} \frac{1}{k} \to \ln(\Delta) \tag{52}$$

Since $\alpha$, $\tau$, and $\Delta > 1$ are constants, $\ln(\Delta)$ is a positive constant. Thus:

$$\mathrm{E}\left[Z_{t_1}^{t_1\Delta}\right] \to \alpha\tau \ln(\Delta) = \Theta(1) \tag{53}$$

This satisfies the condition for scale-invariant total attention in expectation. □

**Lemma 2.** *Consider a set of random variables, $\{L_t\}$, representing attention logits. Let $\tau > 0$ be a lengthscale parameter, and $\beta > 0$ a multiplicative constant. If the attention logits satisfy,*

$$\mathrm{E}\left[\tilde{A}_t \log \tilde{A}_t\right] = \frac{\beta}{t/\tau+1}, \tag{13}$$

*then we have scale-invariant unnormalised attention sparsity (Def. 3.2).*

*Proof.* We want to show that $\mathrm{E}\left[\tilde{N}_{t_1}^{t_1\Delta}\right] = \Theta(1)$ as $t_1 \to \infty$. By definition and linearity of expectation:

$$\mathrm{E}\left[\tilde{N}_{t_1}^{t_1\Delta}\right] = \mathrm{E}\left[\sum_{t=t_1}^{t_1\Delta-1} \tilde{A}_t \log \tilde{A}_t\right] = \sum_{t=t_1}^{t_1\Delta-1} \mathrm{E}\left[\tilde{A}_t \log \tilde{A}_t\right] \tag{54}$$

Using the given condition $\mathrm{E}\left[\tilde{A}_t \log \tilde{A}_t\right] = \frac{\beta}{t/\tau+1}$:

$$\mathrm{E}\left[\tilde{N}_{t_1}^{t_1\Delta}\right] = \sum_{t=t_1}^{t_1\Delta-1} \frac{\beta}{t/\tau+1} = \beta\tau \sum_{t=t_1}^{t_1\Delta-1} \frac{1}{t+\tau} \tag{55}$$

This sum is exactly the same form as in the proof of Lemma 1, just with $\beta$ instead of $\alpha$. Following the same steps using the integral bounds for the harmonic sum derived in Appendix E:

$$\sum_{k=t_1+\tau}^{t_1\Delta-1+\tau} \frac{1}{k} \to \ln(\Delta) \quad \text{as} \quad t_1 \to \infty \tag{56}$$

Therefore, as $t_1 \to \infty$:

$$\mathrm{E}\left[\tilde{N}_{t_1}^{t_1\Delta}\right] \to \beta\tau \ln(\Delta) = \Theta(1) \tag{57}$$

This satisfies the condition for scale-invariant unnormalised attention sparsity in expectation. □

# E    Bounds for Harmonic Sums

For the proofs of Lemma 1 and Lemma 2, we need bounds on the partial harmonic sum $S = \sum_{k=a}^{b} \frac{1}{k}$, where $a = t_1 + \tau$ and $b = t_1\Delta - 1 + \tau$. We can obtain these bounds by comparing the sum to the integral of $f(x) = 1/x$.

Since $f(x) = 1/x$ is a decreasing function for $x > 0$, we have,

$$\int_a^{b+1} \frac{1}{x} dx \leq \sum_{k=a}^{b} \frac{1}{k} \leq \int_{a-1}^{b} \frac{1}{x} dx \tag{58}$$

Evaluating the integrals gives,

$$\ln\left(\frac{b+1}{a}\right) \leq \sum_{k=a}^{b} \frac{1}{k} \leq \ln\left(\frac{b}{a-1}\right). \tag{59}$$

# F    Proof of Theorem 1

**Theorem 1.** *Suppose attention logits $\{L_t\}$ are marginally Gaussian with mean $m_t$ and standard deviation $a_t$ defined by Eq. (16). Assuming $\alpha, \beta, \tau > 0$, $\beta \geq \alpha \log \alpha$, then we have scale-invariant total attention and scale-invariant unnormalised attention sparsity.*

*Proof.* We need to show that the given conditions are sufficient for scale-invariant total attention (Definition 3.1) and scale-invariant unnormalised attention sparsity (Definition 3.2).

**1. Scale-invariant total attention:** We need to show that $\mathrm{E}\left[Z_{t_1}^{t_1\Delta}\right] = \Theta(1)$ as $t_1 \to \infty$. By Lemma 1, this holds if $\mathrm{E}\left[\tilde{A}_t\right] = \frac{\alpha}{t/\tau+1}$. Since $L_t \sim \mathcal{N}(m_t, a_t^2)$, the unnormalised attention weight $\tilde{A}_t = e^{L_t}$ follows a log-normal distribution. The expectation of a log-normal variable $e^X$ where $X \sim \mathcal{N}(\mu, \sigma^2)$ is $e^{\mu+\sigma^2/2}$. Therefore,

$$\mathrm{E}\left[\tilde{A}_t\right] = \mathrm{E}\left[e^{L_t}\right] = e^{m_t + a_t^2/2} \tag{60}$$

Substituting the given expressions for $m_t = -a_t^2 + \beta/\alpha$ and $a_t^2 = 2[\log(t/\tau + 1) - \log\alpha + \beta/\alpha]$:

$$\mathrm{E}\left[\tilde{A}_t\right] = e^{(-a_t^2 + \beta/\alpha) + a_t^2/2} \tag{61}$$

$$= e^{-a_t^2/2 + \beta/\alpha} \tag{62}$$

$$= e^{-[\log(t/\tau+1) - \log\alpha + \beta/\alpha] + \beta/\alpha} \tag{63}$$

$$= e^{-\log(t/\tau+1) + \log\alpha - \beta/\alpha + \beta/\alpha} \tag{64}$$

$$= e^{\log\alpha - \log(t/\tau+1)} \tag{65}$$

$$= e^{\log\left(\frac{\alpha}{t/\tau+1}\right)} \tag{66}$$

$$= \frac{\alpha}{t/\tau + 1} \tag{67}$$

Since this condition matches the requirement of Lemma 1, scale-invariant total attention holds in expectation. The condition $\beta \geq \alpha \log \alpha$ ensures $a_t^2 \geq 0$ for all $t \geq 0$.

**2. Scale-invariant unnormalised attention sparsity:** We need to show that $\mathrm{E}\left[\tilde{N}_{t_1}^{t_1\Delta}\right] = \Theta(1)$ as $t_1 \to \infty$. By Lemma 2, this holds if $\mathrm{E}\left[\tilde{A}_t \log \tilde{A}_t\right] = \frac{\beta}{t/\tau+1}$. We need to compute $\mathrm{E}\left[\tilde{A}_t \log \tilde{A}_t\right] = \mathrm{E}\left[L_t e^{L_t}\right]$. Using the formula for $\mathrm{E}\left[Xe^X\right]$ where $X \sim \mathcal{N}(\mu, \sigma^2)$ from Appendix B, which is $(\mu + \sigma^2)e^{\mu+\sigma^2/2}$:

$$\mathrm{E}\left[L_t e^{L_t}\right] = (m_t + a_t^2)e^{m_t + a_t^2/2} \tag{68}$$

Substitute $m_t = -a_t^2 + \beta/\alpha$:

$$\mathrm{E}\left[L_t e^{L_t}\right] = (-a_t^2 + \beta/\alpha + a_t^2)e^{m_t + a_t^2/2} \tag{69}$$

$$= (\beta/\alpha)e^{m_t + a_t^2/2} \tag{70}$$

From the previous step, we know $e^{m_t + a_t^2/2} = \frac{\alpha}{t/\tau + 1}$. Substituting this:

$$\mathrm{E}\left[L_t e^{L_t}\right] = (\beta/\alpha)\left(\frac{\alpha}{t/\tau + 1}\right) \tag{71}$$

$$= \frac{\beta}{t/\tau + 1} \tag{72}$$

Since this condition matches the requirement of Lemma 2, scale-invariant unnormalised attention sparsity holds.

Therefore, under the given conditions, we have both scale-invariant total attention and scale-invariant unnormalised attention sparsity. $\qquad\square$

## G   Further experimental details

We provide experimental details here, and we provide the code used in the supplementary materials.

### G.1   Pretraining from scratch

Our base model is a `modded-nanogpt` (Jordan et al., 2024a) variant, which is similar to GPT-2 (Radford et al., 2019), but has the following main differences from GPT-2: RMSNorm for layer normalization (applied before the attention and MLP blocks, as well as to the token embeddings and the final output layer), squared ReLU activations, and QK Normalization (RMSNorm on query and key projections). All models used a vocabulary size of 50304, with text tokenized with the GPT-2 tokenizer (Wolf et al., 2020). We implemented scale-invariant attention and ALiBi using FlexAttention (Dong et al., 2024).

**162M parameter model.**   The smaller model had 12 layers, 768 hidden dimension, and 6 heads. We optimized embedding parameters using Adam with learning rate $\gamma = 0.3$, $\beta = (0.9, 0.95)$. We optimized linear layers with Muon, with no weight decay, $\gamma = 0.02$ and momentum $0.95$. For remaining parameters (unembedding and LogN scalings, if LogN trick was active), we optimized with Adam, using $\gamma = 0.002$, $\beta = (0.9, 0.95)$. We trained for 4578 steps. The batch size was $8 \times 65536/L_{\mathrm{tr}}$, with more gradient accumulation for shorter training lengths. We scheduled the learning rate with a linear schedule for all parameters, with no warmup, constant learning rate for 3270 steps, and linear cooldown for the remaining 1308 steps. We vary the training context length when pretraining with this model, from 4096(4k), to 16384(16k), and 65536(64k). We validate at 4k, 16k, and 64k sequence lengths every 125 steps.

**304M parameter model.**   For the larger model, we used the same settings as the smaller model, but with the following changes. The larger model had 16 layers, 1024 hidden dimension, and 8 heads. We trained the model for 10900 steps, processing approximately 10B tokens. We trained for 2 accumulation steps on 4 GPUs, with 28 sequences per batch. We scaled learning rates following $\mu$Param (Wortsman et al., 2023), which involves multiplying learning rates of the linear layers by $768/1024$, to adjust for changing the model width. Learning rates of other layers (embedding, unembedding, and LogN scalings if active) were not changed. We used a cosine learning rate schedule, with no warmup, and a minimum learning rate of 0 (at the end of training). All runs on the 304M parameter models were with 4096 training sequence length. We validated at 4k, 16k, 64k sequence lengths every 250 steps.

**RoPE hyperparameters.**   We used a base $\theta$ of $10,000$ for RoPE, and an effective base of $1024$ for the angular frequencies in $p$-RoPE. For RoPE+NTK scaling (note the scaling applies only when the inference sequence length is longer than the training sequence length), we scaled $\theta$ by $(L_{\mathrm{inf}}/L_{\mathrm{tr}})^{d/(d-2)}$, where $L_{\mathrm{tr}}$ is the training sequence length, $L_{\mathrm{inf}}$ is the inference sequence length, and $d$ is the RoPE head dimension (128 for both models).

**Dataset.** We used subsets of FineWeb (Penedo et al., 2024) for pretraining from scratch: a 10B token subset for the 162M model, and a 100B token subset for the 304M model (from which ∼10B tokens were used for its training). We keep the training data ordering fixed, and we keep the validation set identical across all runs to reduce variance.

**Seeds.** We repeat the experiment for 3 different seeds when training at 4k on the 162M parameter model. We train with 1 seed at 4k on the 304M parameter model, and at 16k/64k on the 304M parameter model due to compute limitations.

**Compute.** We trained the smaller (162M) models on single A100 80G GPUs. We trained the 304M models on 4xH100 grace hopper nodes using distributed data parallelism.

### G.2 Needle-in-a-haystack

The Needle in a Haystack experiments were conducted by fine-tuning the pre-trained 162M parameter models. We fine-tune with the same learning rate as above, but with 100 warmup, 100 constant, and 100 warmdown steps (decaying to $\gamma = 0$). We use the same optimizers as in the pretraining phase. We train for 300 steps on sequences of length 4096, and batch size 8 with 8 accumulation steps. We repeated with 3 seeds.

The task is to generate responses of the form "`city1=needle1;city2=needle2;city3=needle3`", where the cities and needles are embedded uniformly at random into samples from C4 in the form "`The special magic <city> number is <7_digit_number>.`". We sample several times from the C4 dataset (concatenating samples) and remove tokens until we have the necessary number of tokens — 4096(4k) for training, and 4096(4k), 16384(16k), 65536(64k) for validation. Only the expected response tokens are included in the loss (the prompt/context tokens are masked). We repeat the task for three different seeds. The validation accuracies presenting in Table 1 are calculated by the proportion of times that cities *and* numbers are output correctly.

### G.3 Resources required to reproduce experiments

To reproduce results, 80G GPUs are required. We used 80G A100s, and 80G H100 grace hopper nodes. In terms of time taken to execute each experiment type, we give estimates of the resources required:

- pretraining 162M parameter model w/o flex attention takes roughly 4/8/22 A100 hours at 4k/16k/64k;
- pretraining 162M parameter model with flex attention takes roughly 8/16/44 A100 hours at 4k/16k/64k;
- pretraining 304M parameter model takes roughly 36 4xH100 node hours;
- fine-tuning on needle-in-a-haystack task takes roughly 3 A100 hours.

To obtain our results, each experiment is executed several times. In particular, needle-in-a-haystack with 3 seeds per method, pretraining at 4k with 162M model is 3 seeds per positional encoding method. The remaining experiments are executed once per positional encoding method. This gives a total of ∼ 100 4xH100 hours, ∼ 550 1xA100 hours. We estimate that very roughly that amount again was spent configuring the experiments correctly, and on preliminary/failed experiments.

## H   The optimal lengthscale, $\tau$, is around 10

By introducing scale-invariant attention, we introduce one extra lengthscale hyperparameter, $\tau$, which represents the size of the 'chunks' we attend over. To select $\tau$, we trained at 4k for $\tau \in \{10^{-2}, 10^{-1}, 10^{0}, 10^{1}, 10^{2}\}$, and compared validation losses (shown in Fig. 5). Validation performance is very similar amongst different $\tau$ at the training context length (though $\tau = 10$ is strictly best), but as we extend to out of distribution context lengths (64k, 16× the training context length) the benefit of $\tau = 10$ becomes clearer.

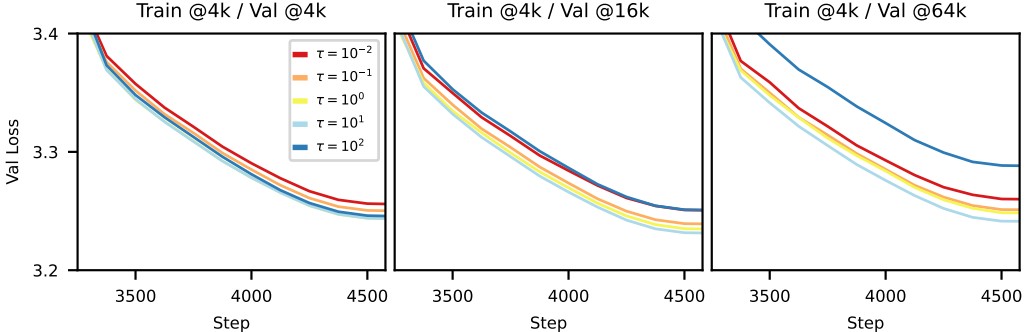

Figure 5: Validation losses at 4k (left), 16k (middle), and 64k (right) context lengths, for a GPT-2-like model trained with scale-invariant attention for varying $\tau$. The models were trained at 4k context length.

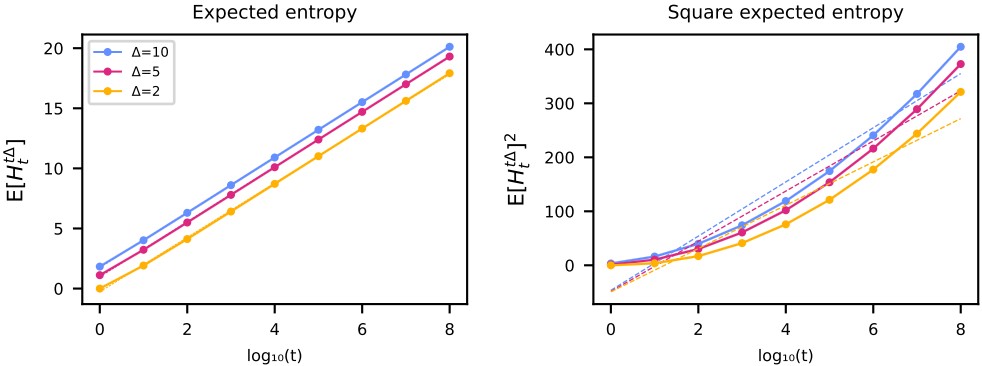

Figure 6: Scaling of expected entropy-in-range for standard attention with an independent Gaussian assumption on the logits. We calculate expected entropies in the range $[t, t\Delta]$ for different $t$ and $\Delta \in \{2, 5, 10\}$, with the lengthscale $\tau$ set to 10. The dashed lines show the best linear fit of the data. We empirically find that standard Gaussian logits give logarithmic entropy (left panel).

# I  Extra Results

## I.1  Entropy scaling of regular attention

In the main text, we illustrated in Fig. 2 that scale-invariant attention method under a Gaussian assumption has sub-logarithmic expected entropy. Fig. 6 empirically shows that expected entropy in a range $t$ to $t\Delta$ for standard/unscaled attention instead scales logarithmically with $t$.

## I.2  Pretrain @16k

For completeness, we include validation losses when training at 16k in Fig. 7. We see again that scale-invariant $p$-RoPE outperforms other methods over a range of validation lengths, with the improvements becoming more noticable at the longest validation length of 64k.

## I.3  Alternative scale-invariant attention variants

In Section 3 we presented scale-invariant $p$-RoPE as our proposed method. In preliminary experiments however, it was very natural to also consider scale-invariant RoPE and scale-invariant NoPE. We show results when training at 4k in Fig. 8. Scale-invariant RoPE performs almost as well as scale-invariant $p$-RoPE when evaluating at the training context length, but underperforms more as we move to 16k and 64k. On the other hand, scale-invariant NoPE underperforms scale-invariant $p$-RoPE, yet generalises

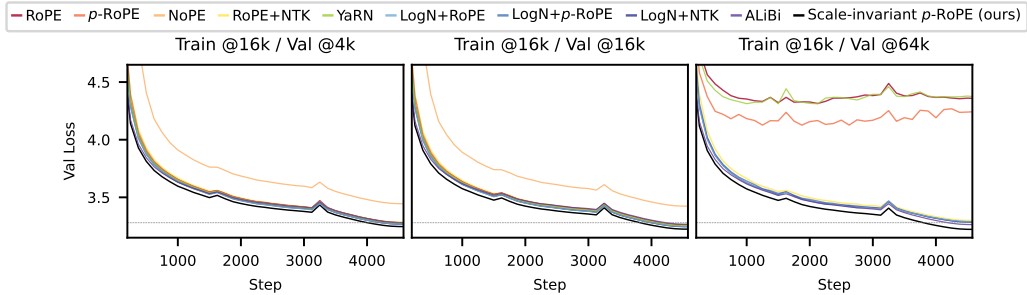

Figure 7: Validation losses at 4k / 16k / 64k context lengths, for a 162M parameter GPT-2-style model trained at 16k. NoPE omitted on the right-most plot to avoid excessive zooming.

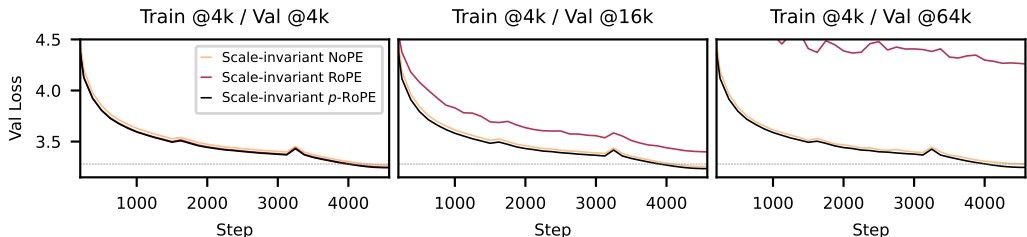

Figure 8: Validation losses at 4k / 16k / 64k of scale-invariant NoPE, RoPE, and $p$-RoPE, for a 162M parameter GPT-2-style model trained at 4k .

to long contexts. We hypothesised that scale-invariant RoPE does not long-context generalise due to RoPE's low frequences (i.e. high wavelengths) interfering with the position-dependent scale-invariant transformation, $L_t \mapsto a_t L_t + m_t$.

### I.4 Infini-Attention

During the review process, we were asked to compare scale-invariant attention to Infini-attention (Munkhdalai et al., 2024). Since infini-attention is a method for compressing the KV-cache, it is slightly different to the other methods we compare to (which primarily control entropy), and so we include these results separately. See Table 3, which shows that while infini-attention gives long-context generalization, it is not as strong as our method.

### I.5 Larger models

We also investigated the ability for scale-invariant attention to generalise to longer contexts in larger models by continual pretraining Llama 2 7B (Touvron et al., 2023). We chose Llama 2 for this experiment because it is one of the few models that have not already mid-trained at a longer context length.

Specifically, we continually pretrained at 4k context length after replacing the default attention mechanism with various other methods, and we looked at validation loss at out-of-distribution lengths (4k/16k/64k). We trained using the Torchtune (2024) library, using data from FineWeb (Penedo et al., 2024), with AdamW and a learning rate of $2 \times 10^{-5}$.

We see in Table 4 that changing the attention mechanism degraded the loss at the training context length, but the performance of scale-invariant $p$-RoPE far exceeds the other methods at 16k and 64k.

Table 3: Final mean validation losses ($\pm 1$ standard error across 3 seeds) for different methods when training with 4k context length on a 162M parameter GPT-2-style model.

| Method | Val @ 4k | Val @ 16k | Val @ 64k |
|---|---|---|---|
| RoPE | $3.261 \pm 0.001$ | $3.936 \pm 0.010$ | $5.260 \pm 0.014$ |
| Scale-invariant $p$-RoPE (ours) | $\mathbf{3.244} \pm 0.001$ | $\mathbf{3.235} \pm 0.001$ | $\mathbf{3.247} \pm 0.001$ |
| Infini-RoPE | $3.296 \pm 0.003$ | $3.302 \pm 0.004$ | $3.310 \pm 0.008$ |
| Infini-$p$-RoPE | $3.295 \pm 0.003$ | $3.303 \pm 0.005$ | $3.311 \pm 0.009$ |

Table 4: Validation performance when fine-tuning Llama-2 7B with different attention methods on $\sim$50M tokens. RoPE and RoPE+NTK (denoted *) were not fine-tuned.

| Method | Val loss @4k | Val loss @16k | Val loss @64k |
|---|---|---|---|
| RoPE* | 1.968 | 7.036 | 8.815 |
| $p$-RoPE | 2.029 | 3.504 | 6.800 |
| NoPE | 4.152 | 6.172 | 7.730 |
| RoPE+NTK* | 1.988 | 3.090 | 7.523 |
| YaRN | **1.957** | 7.004 | 8.763 |
| LogN+RoPE | 1.966 | 6.932 | 8.726 |
| LogN+$p$-RoPE | 2.049 | 2.984 | 6.224 |
| LogN+NTK | 1.966 | 3.063 | 7.413 |
| ALiBi | 2.750 | 2.745 | 2.744 |
| Scale-invariant $p$-RoPE (ours) | 2.163 | **2.193** | **2.252** |

# J Checking Gaussianity of attention logits

In our analysis in Section 3.2 we assume that the unmodified logits (query-key products), $\bar{L}_t$'s, are standard Gaussian. In this section, we empirically verify that the logits are Gaussian by looking at QQ-plots.

We consider several sizes of model, including 1B and 8B Llama variants (Grattafiori et al., 2024), and Gemma 2 27B Team et al. (2024). We calculate logits with the introduction paragraph of a Wikipedia page [1] as the input. In Figs. 9,10,11,12,13,14 we show quantiles of $\{\bar{L}_t\}_{t>0}$'s (i.e. lower triangular part of the $QK^T$ matrix) for each layer, aggregated over the input and the attention heads in each layer. Note that we do not aggregate over the 'beginning of sequence' token, as it is an outlier attention sink (Gu et al., 2024).

---

[1] https://en.wikipedia.org/wiki/New_England

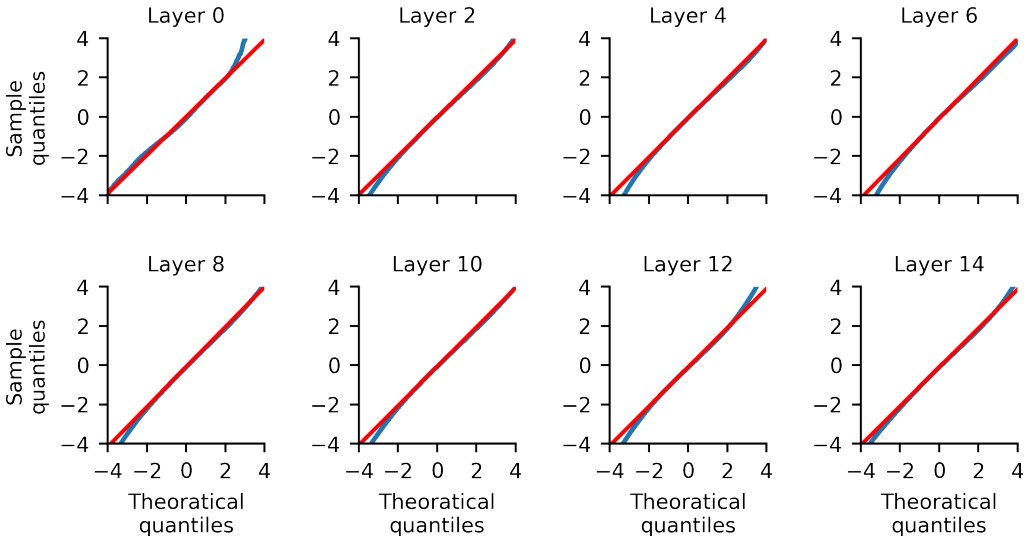

Figure 9: Quantile-Quantile (QQ) plots of attention logits (without RoPE applied) in a Llama-1B model (blue line), with theoretical quantiles of a Gaussian shown by the red line.

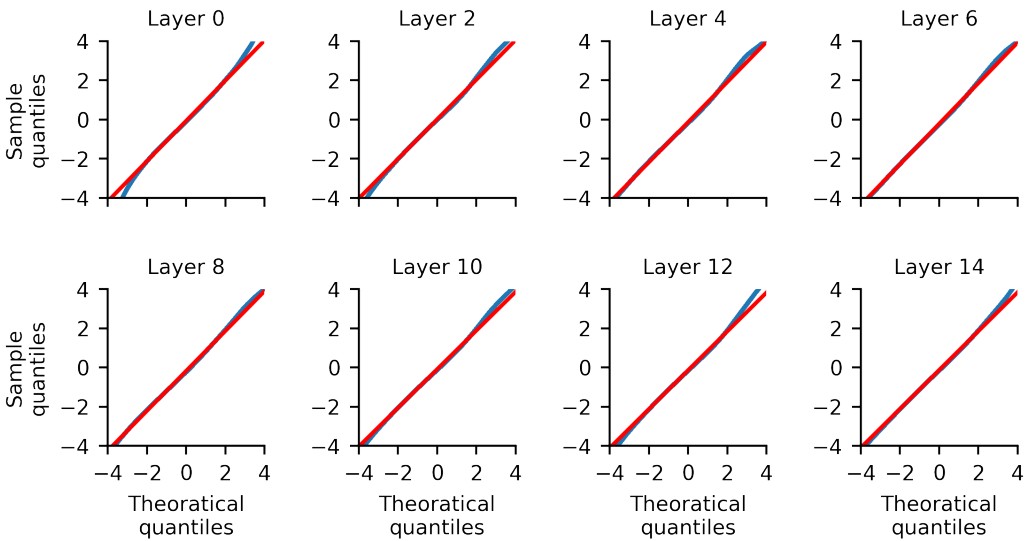

Figure 10: Quantile-Quantile (QQ) plots of attention logits (with RoPE applied) in a Llama-1B model (blue line), with theoretical quantiles of a Gaussian shown by the red line.

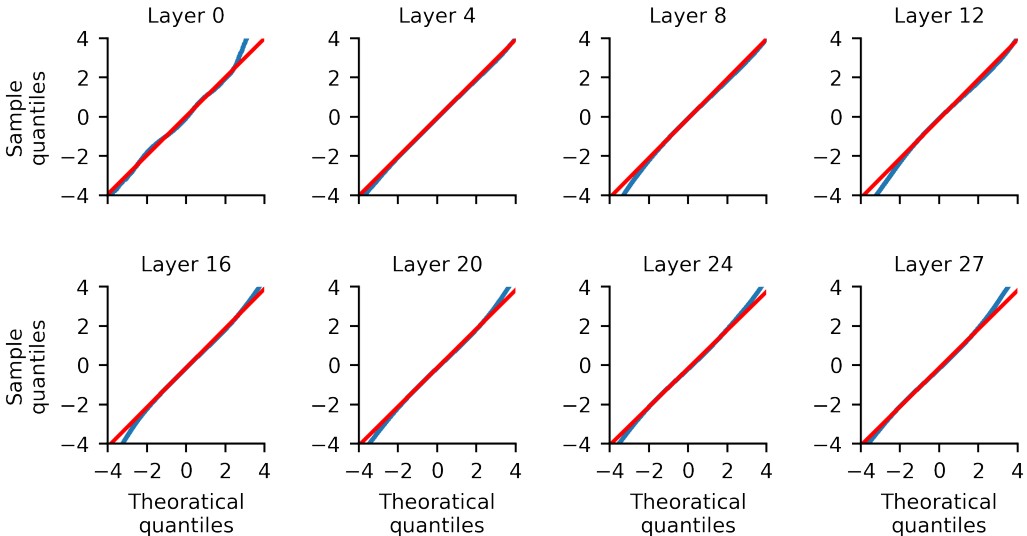

Figure 11: Quantile-Quantile (QQ) plots of attention logits (without RoPE applied) in a Llama-8B model (blue line), with theoretical quantiles of a Gaussian shown by the red line.

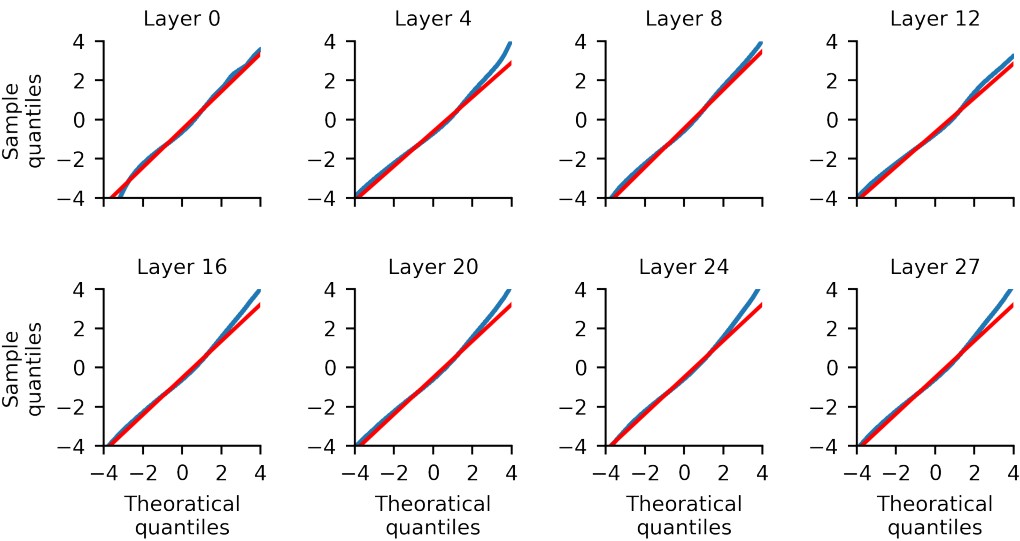

Figure 12: Quantile-Quantile (QQ) plots of attention logits (with RoPE applied) in a Llama-8B model (blue line), with theoretical quantiles of a Gaussian shown by the red line.

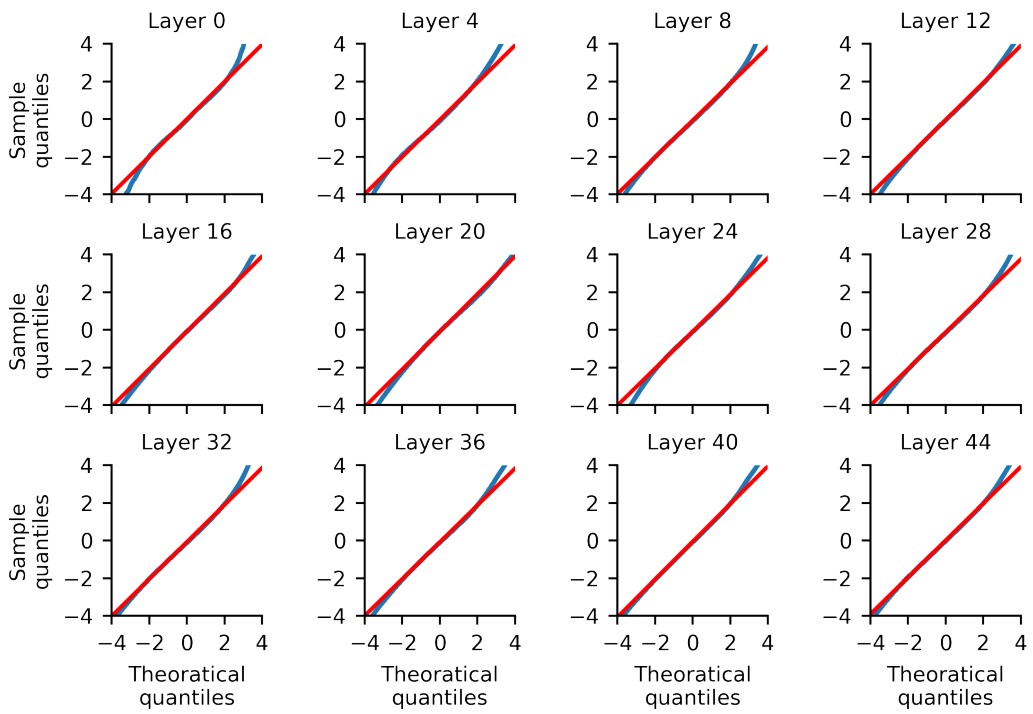

Figure 13: Quantile-Quantile (QQ) plots of attention logits (without RoPE applied) in a Gemma 2 27B model (blue line), with theoretical quantiles of a Gaussian shown by the red line.

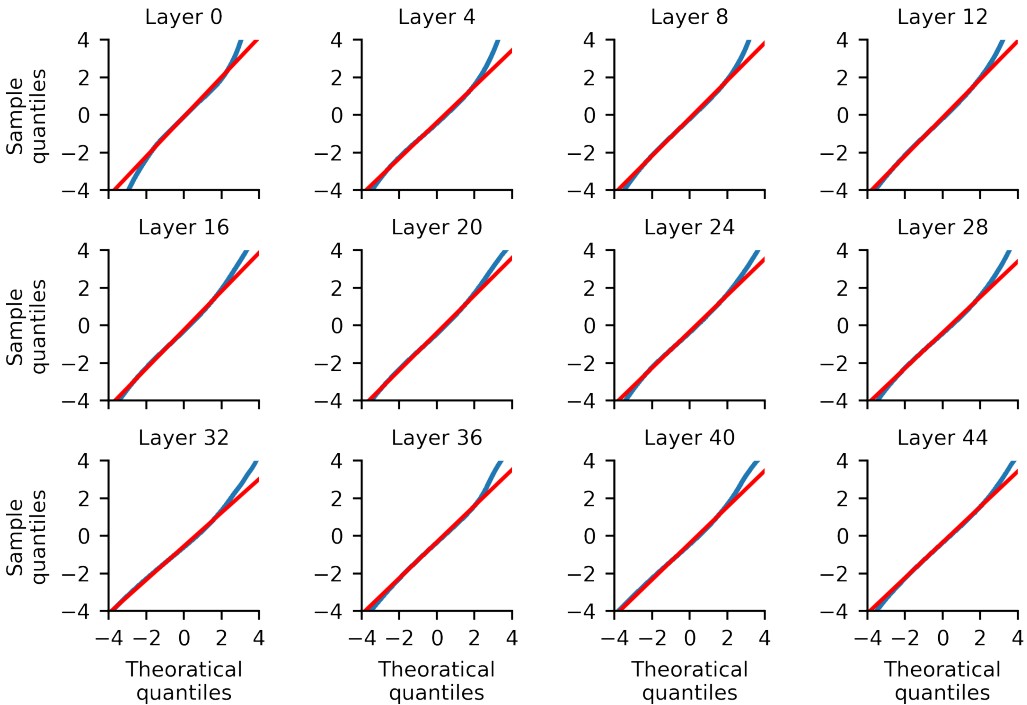

Figure 14: Quantile-Quantile (QQ) plots of attention logits (with RoPE applied) in a Gemma 2 27B model (blue line), with theoretical quantiles of a Gaussian shown by the red line.

# K Licenses

- This project uses a modified version of modded-nanogpt, https://github.com/KellerJordan/modded-nanogpt which is MIT licensed.
- This project uses a 10B subset of the fineweb dataset, https://huggingface.co/datasets/kjj0/fineweb10B-gpt2, which is MIT licensed.
- This project uses a 100B subset of the fineweb dataset, https://huggingface.co/datasets/kjj0/fineweb100B-gpt2, which is MIT licensed.
- This project uses the C4 dataset, https://huggingface.co/datasets/kjj0/fineweb100B-gpt2, which licensed under the Open Civic Data Attribution License (OCD-BY).

