# OpenReview forum: "Scale-invariant attention"
_NeurIPS.cc/2025/Conference — NeurIPS 2025 poster_

### Official Review · Reviewer_tNeo · 2025-06-28

**Clarity:** 2
**Significance:** 3
**Originality:** 2
**Rating:** 4
**Confidence:** 4

**Summary:**

This paper addresses the challenge of long-context generalization in LLMs by proposing scale-invariant attention, which satisfies two key properties: scale-invariant total attention (the expected sum of unnormalized attention weights over any token range \([t, t\Delta)\) is asymptotically constant) and scale-invariant attention sparsity (attention is denser in local ranges and sparser in global ranges, with entropy growing sub-logarithmically). Under a Gaussian assumption, the authors derive a position-dependent transformation of attention logits (\(L_t = a_t \bar{L}_t + m_t\)) to satisfy these properties. Experimental results show that the proposed "scale-invariant p-RoPE" outperforms baselines (e.g., RoPE, LogN, ALiBi) in validation loss across various training/validation context lengths, enables strong zero-shot generalization from short to long contexts, and performs well in the needle-in-a-haystack retrieval task. The main contributions include introducing the two scale-invariant properties, deriving the logit transformation, and validating its effectiveness empirically

**Questions:**

(1) The paper only evaluates models with 162M and 304M parameters, but contemporary LLMs often have billions of parameters. How does scale-invariant attention perform on larger models (e.g., 10B+ parameters)? Larger models may exhibit different attention behaviors due to increased capacity; verifying scalability is crucial for assessing real-world applicability. For example, do the hyperparameters (e.g., \(\tau=10\)) remain optimal, or do they need adjustment for larger architectures?

(2) The logit transformation is derived under the assumption that base logits (\(\bar{L}_t\)) are IID Gaussian. However, real attention logits in trained models may deviate from Gaussianity. The paper mentions a QQ plot for Llama-3B, but how robust is the proposed method when logits are non-Gaussian (e.g., skewed or with heavier tails)? Would the transformation still satisfy scale-invariant properties, or is additional adaptation needed?

(3) While the paper compares with several baselines, it does not include some recent long-context methods, such as Infini-Attention or Ring Attention. How does scale-invariant attention compare to these methods in terms of performance, computational efficiency, and generalization? Including such comparisons would better situate the work within the broader landscape of long-context LLMs.

(4) The paper mentions that scale-invariant RoPE underperforms scale-invariant p-RoPE due to RoPE’s low-frequency components interfering with the transformation. However, it does not elaborate on why low-frequency components cause this interference. Could this be mitigated by modifying RoPE’s frequency spectrum (e.g., adjusting the base \(\theta\))? A deeper analysis of this interaction would clarify the choice of p-RoPE as the base.

(5) The needle-in-a-haystack task uses fine-tuning, which is unusual (most studies use prompting). The paper states that prompting alone was insufficient, but it does not explain why (e.g., model size, pretraining data). Would larger models or more pretraining enable the proposed method to perform this task with prompting? This information is important for assessing its practical utility in retrieval scenarios without fine-tuning.

**Ethical Concerns:**

["NO or VERY MINOR ethics concerns only"]

**Final Justification:**

The response from authors address my major concerns. So i raise the score.

**Limitations:**

yes

**Quality:**

2

**Strengths And Weaknesses:**

## Strength
1. The paper rigorously defines scale-invariant total attention and sparsity, provides mathematical proofs (Lemmas 1, 2, and Theorem 1) for the conditions under which these properties hold, and derives a concrete logit transformation, ensuring theoretical soundness.
2. The proposed method involves a simple position-dependent transformation of attention logits, with only one hyperparameter (\(\tau\)) after simplification, making it easy to implement and integrate with existing frameworks like p-RoPE.
3. The paper effectively distinguishes its approach from related methods (e.g., LogN’s position-independent scaling vs. its position-dependent transformation), highlighting its unique ability to balance local and global attention.

## Weakness
1. The paper only validates the proposed method on models with 162M and 304M parameters, like GPT2, while state-of-the-art LLMs typically have billions of parameters. The authors acknowledge this limitation, noting that resource constraints prevented testing on larger models, which raises questions about whether the conclusions generalize to commercially deployed large-scale models.
2. Although the paper compares with methods like RoPE, LogN, and ALiBi, it omits comparisons with several recent long-context attention mechanisms (e.g., Infini-Attention, Ring Attention) that also aim to address length generalization. This gap makes it harder to situate the proposed method within the broader landscape of current long-context LLM research .
3. The theoretical derivation of the scale-invariant attention transformation hinges on the assumption that base attention logits (\(\bar{L}_t\)) follow an IID Gaussian distribution. While the paper includes a QQ plot for Llama-3B to partially verify this, it lacks in-depth analysis of how the method performs when logits deviate significantly from Gaussianity (e.g., skewed distributions or heavy tails), which could limit its robustness in real-world scenarios .
4. The issue that this paper aims to address is significant, while the presentation and experimental desmonstration is poor for me. How the proposed method motivated by studying at entropy? how it affect the design? and why this factor not evaluated at experiments?

---

> ### Author Rebuttal · Authors · 2025-07-30
>
> Thanks for your careful and positive review!
>
> > How the proposed method motivated by studying at entropy? how it affect the design? and why this factor not evaluated at experiments?
>
> As discussed in Sec 3.1, we used the entropy of the distribution defined by the attention weights to measure sparsity.  For a sequence of length $T$, if the entropy is $\log T$, then the attention must be uniformly distributed across all tokens.  In contrast, if the entropy is $0$, then the attention must be focused on a single token.  Outside of these extreme cases, an entropy of $\log n$ could be generated by an attention weight of $1/n$ assigned to $n$ tokens and an attention weight of $0$ assigned to all the other tokens (though of course in practice many other distributions will also have an entropy of $\log n$).
>
> We evaluated the entropy in Figure 1.
>
>
>
> > For example, do the hyperparameters (e.g., ($\tau=10$)) remain optimal, or do they need adjustment for larger architectures?
>
> We agree that we established the optimal value of $\tau$ experimentally in a particular setup.
> Thus we would recommend that anyone using scale-invariant attention should check that $\tau=10$ is close-to-optimal in their setting.
> Given our intuition that $\tau$ measures the "size of 'chunks' we attend over", we would not expect the optimal $\tau$ to be too far from $10$.
>
>
>
> > (1) The paper only evaluates models with 162M and 304M parameters, but contemporary LLMs often have billions of parameters. How does scale-invariant attention perform on larger models (e.g., 10B+ parameters)? Larger models may exhibit different attention behaviors due to increased capacity; verifying scalability is crucial for assessing real-world applicability. For example, do the hyperparameters (e.g., ($\tau=10$)) remain optimal, or do they need adjustment for larger architectures?
>
> We have added further experiments --- continual pretraining on Llama 2 7B at 4k context length, and evaluating the validation loss at longer context sizes (see table below) to demonstrate that scale-invariant attention extends to larger scale models.
>
>
> | Method                          | Val loss @ 4k | Val loss @ 16k | Val loss @ 64k |
> |--------------------------------|----------|-----------|-----------|
> | RoPE*                          | 1.968    | 7.036     | 8.815     |
> | $p$-RoPE                       | 2.029    | 3.504     | 6.800     |
> | NoPE                           | 4.152    | 6.172     | 7.730     |
> | RoPE+NTK*                      | 1.988    | 3.090     | 7.523     |
> | YaRN                           | 1.957    | 7.004     | 8.763     |
> | LogN+RoPE                      | 1.966    | 6.932     | 8.726     |
> | LogN+$p$-RoPE                  | 2.049    | 2.984     | 6.224     |
> | LogN+NTK                       | 1.966    | 3.063     | 7.413     |
> | ALiBi                          | 2.750    | 2.745     | 2.744     |
> | Scale-invariant $p$-RoPE (ours)| 2.163    | 2.193     | 2.252     |
>
> We found that while replacing the attention mechanism had adverse effects @ 4k, scale-invariant attention yielded by far the best results OOD @ 64k. (Note that we chose Llama 2 for this continual pretraining experiment because it is one of the few models that have not already mid-trained at a longer context length).
>
>
> > (2) The logit transformation is derived under the assumption that base logits ($\bar{L}_t$) are Gaussian. However, real attention logits in trained models may deviate from Gaussianity. The paper mentions a QQ plot for Llama-3B, but how robust is the proposed method when logits are non-Gaussian (e.g., skewed or with heavier tails)? Would the transformation still satisfy scale-invariant properties, or is additional adaptation needed?
>
> To confirm that the distribution over logits is approximately Gaussian, we have added additional QQ plots for both larger Llama-3 models as well as Gemma models. Given that we were not able to find a realistic setting where the logits deviate from Gaussianity, we were not able to study this possibility (and indeed, it does seem that it would be rare in practice).
>
> > (3) While the paper compares with several baselines, it does not include some recent long-context methods, such as Infini-Attention or Ring Attention. How does scale-invariant attention compare to these methods in terms of performance, computational efficiency, and generalization? Including such comparisons would better situate the work within the broader landscape of long-context LLMs.
>
> Infini-attention is a method for compressing the KV-cache, and is thus slightly different to the other methods we compare to (which primarily control entropy). We nonetheless ran pretraining experiments with Infini-attention, and found that while Infini-attention shows length generalization, it is not as strong as our method. See the table below; we include standard attention (RoPE) for comparison,
>
> | Method                         | Val @ 4k         | Val @ 16k        | Val @ 64k        |
> |-------------------------------|------------------|------------------|------------------|
> | RoPE                          | 3.261 ± 0.001    | 3.936 ± 0.010    | 5.260 ± 0.014    |
> | **Scale-invariant *p*-RoPE** (ours) | **3.244 ± 0.001** | **3.235 ± 0.001** | **3.247 ± 0.001** |
> |-------------------------------|------------------|------------------|------------------|
> | Infini+RoPE                   | 3.296 ± 0.003    | 3.302 ± 0.005    | 3.310 ± 0.010    |
> | Infini+*p*-RoPE               | 3.295 ± 0.003    | 3.303 ± 0.006    | 3.311 ± 0.011    |
>
> Ring Attention is a method for distributing the attention computation more efficiently over GPUs/TPUs --- it doesn't actually alter the attention computation at all. Therefore it will not improve long context performance, when coupled e.g. with RoPE, when out of the training distribution.
>
> > (4) The paper mentions that scale-invariant RoPE underperforms scale-invariant p-RoPE due to RoPE’s low-frequency components interfering with the transformation. However, it does not elaborate on why low-frequency components cause this interference. Could this be mitigated by modifying RoPE’s frequency spectrum (e.g., adjusting the base ($\theta$))? A deeper analysis of this interaction would clarify the choice of p-RoPE as the base.
>
> It is understood that $p$-RoPE tends to work better than RoPE in the long-context settings by making semantic attention patterns more robust at long distances [4]. Indeed both DeepSeek and Kimi implement a form of $p$-RoPE via MLA: see in MLA that RoPE is not applied to the compressed keys and queries [5, section 2.1.1]. Further, increasing the base $\theta$, which is common practice, can be seen as approximating $p$-RoPE. We therefore believe our results on RoPE vs $p$-RoPE are simply in keeping with these more general, and well understood, trends.
>
>
> > (5) The needle-in-a-haystack task uses fine-tuning, which is unusual (most studies use prompting). The paper states that prompting alone was insufficient, but it does not explain why (e.g., model size, pretraining data). Would larger models or more pretraining enable the proposed method to perform this task with prompting? This information is important for assessing its practical utility in retrieval scenarios without fine-tuning.
>
> Scale-invariant attention makes a relatively minor change to the attention mechanism that enables zero-shot length generalisation.  When keeping the context size at or below the training context size, we'd expect scale-invariant models to perform in almost exactly the same way as standard attention, and that is bourne out in our results (see Val @ 4k columns in Table 1 and Table 2). We found that the networks we pretrained with both standard attention and scale-invariant attention were unable to perform the needle-in-a-haystack task with just prompting, likely due to the small size of the models.  If we were to train a larger model on more tokens, we would expect to be able to get needle-in-a-haystack by prompting with both standard attention and scale-invariant attention.  Further, in the paper, every task we tried showed zero-shot length generalisation for scale-invariant attention and not for standard attention (Table 1, Table2, Figure 3); it is reasonable to expect that pattern to continue for prompted needle-in-a-haystack and on a larger model.
>
>
> [1]: Zhou, Yang, et al. "GSM-Infinite: How Do Your LLMs Behave over Infinitely Increasing Context Length and Reasoning Complexity?." arXiv preprint arXiv:2502.05252 (2025).
>
> [2]: Hong, K., Troynikov, A., & Huber, J. (2025, July). Context Rot: How Increasing Input Tokens Impacts LLM Performance. Chroma Research.
>
> [3]: Li, Tianle, et al. "Long-context llms struggle with long in-context learning." arXiv preprint arXiv:2404.02060 (2024).
>
> [4]: Barbero et al., Round and round we go! what makes rotary positional encodings useful? ICLR
>
> [5]: Liu, Aixin, et al. "Deepseek-v3 technical report." arXiv.

---

> > ### Comment · Reviewer_tNeo · 2025-08-04
> >
> > Thank you. i have no further questions.

---

> > > ### Author Response · Authors · 2025-08-04
> > > **Response**
> > >
> > > Thanks for considering our response!
> > >
> > > Given that we have been able to address your questions, we would be very grateful if you would consider increasing your score?

---

> > > > ### Comment · Reviewer_tNeo · 2025-08-06
> > > >
> > > > yes, i will increase it.

---

### Official Review · Reviewer_yBmN · 2025-07-01

**Clarity:** 2
**Significance:** 3
**Originality:** 2
**Rating:** 4
**Confidence:** 3

**Summary:**

This paper introduces a new attention mechanism—scale-invariant attention—aimed at improving large language models’ ability to generalize from training on short contexts to inference on long contexts. The authors formalize two desirable properties for attention over long sequences: scale-invariant total attention, which ensures that attention is evenly distributed across different positional ranges, and scale-invariant attention sparsity, which ensures that attention becomes increasingly sparse over more distant contexts. To achieve these properties, the authors propose a simple transformation of the attention logits using position-dependent multiplicative and additive terms. They provide theoretical proofs under Gaussian assumptions that this transformation satisfies both scale-invariant total attention and (weak) scale-invariant attention sparsity.

**Questions:**

1. Can the authors compare their method to memory-based approaches or retrieval-augmented LLMs, particularly on long-context retrieval tasks?
2. Does the position-dependent logit scaling affect optimization dynamics? Empirical data on convergence speed or stability would be helpful.
3. Can the authors provide more detailed empirical or theoretical analysis on why p-RoPE pairs better with scale-invariant attention than RoPE? Frequency spectrum visualizations could help.
4. While the study focuses on GPT-style models, could this method extend to encoder-decoder or encoder-only models like BERT or T5?

**Ethical Concerns:**

["NO or VERY MINOR ethics concerns only"]

**Final Justification:**

I will keep my score positive considering all rebuttal and discussions.

**Limitations:**

yes

**Quality:**

3

**Strengths And Weaknesses:**

**Strengths**
1. The paper provides a solid mathematical foundation for its proposed attention mechanism, with clear definitions and provable guarantees for the desired properties.
2. The proposed logit transformation is lightweight and easy to implement, requiring only position-dependent scaling and biasing of logits.
3. The method shows strong zero-shot performance without requiring long-context pretraining, which has implications for cost-efficient LLM deployment.

**Weaknesses**
1. Experiments are conducted on relatively small-scale models (162M and 304M parameters), which may limit confidence in applicability to larger foundation models.
2. The authors do not analyze whether their logit transformation impacts convergence speed or training stability, which may be useful for practitioners.
3. The three lines drawn in Figure 2 have low differentiation, so it is recommended that the author use different colors to mark them.
4. The performance in Table 2 is slightly lower at 4k. It is recommended that the author analyze the reasons.

---

> ### Author Rebuttal · Authors · 2025-07-30
>
> Thanks for your positive and careful review.
>
> > Experiments are conducted on relatively small-scale models (162M and 304M parameters), which may limit confidence in applicability to larger foundation models.
>
> We have added further experiments --- continual pretraining on Llama 2 7B at 4k context length, and evaluating the validation loss at longer context sizes (see table below) to demonstrate that scale-invariant attention extends to larger scale models.
>
>
> | Method                          | Val loss @ 4k | Val loss @ 16k | Val loss @ 64k |
> |--------------------------------|----------|-----------|-----------|
> | RoPE*                          | 1.968    | 7.036     | 8.815     |
> | $p$-RoPE                       | 2.029    | 3.504     | 6.800     |
> | NoPE                           | 4.152    | 6.172     | 7.730     |
> | RoPE+NTK*                      | 1.988    | 3.090     | 7.523     |
> | YaRN                           | 1.957    | 7.004     | 8.763     |
> | LogN+RoPE                      | 1.966    | 6.932     | 8.726     |
> | LogN+$p$-RoPE                  | 2.049    | 2.984     | 6.224     |
> | LogN+NTK                       | 1.966    | 3.063     | 7.413     |
> | ALiBi                          | 2.750    | 2.745     | 2.744     |
> | Scale-invariant $p$-RoPE (ours)| 2.163    | 2.193     | 2.252     |
>
> We continually pretrained on Llama 2 at 4k context length after replacing the default attention mechanism with various other methods, and we looked at validation loss at out of distribution lengths (4k/16k/64k). Changing the attention mechanism degraded the loss at the training context length, but scale-invariant $p$-RoPE performed best out of distribution. Note that we chose Llama 2 for this continual pretraining experiment because it is one of the few models that have not already mid-trained at a longer context length.
>
> > The authors do not analyze whether their logit transformation impacts convergence speed or training stability, which may be useful for practitioners.
>
> Figures 3, 4, 5, 7, 8 all show training curves (i.e. validation loss vs. number of gradient descent steps), and we can clearly see that number of steps required to reach a given loss is improved by our method, and it does not give rise to any indications of a loss-of-stability.
>
> > The three lines drawn in Figure 2 have low differentiation, so it is recommended that the author use different colors to mark them.
>
> Fixed! Thanks!
>
> > The performance in Table 2 is slightly lower at 4k. It is recommended that the author analyze the reasons.
>
> This difference is statistically insignificant at the 1\% level so all we can say is that the methods are not distinguishable.
>
>
> #### Questions:
>
> > Can the authors compare their method to memory-based approaches or retrieval-augmented LLMs, particularly on long-context retrieval tasks?
>
> Long context (LC) methods and RAG methods are very different methods with very different performance characteristics.  For instance, RAG is thought to perform very poorly for code [1], and LC is generally thought to outperform RAG for QA benchmarks, especially for Wikipedia based tasks [2]. Though of course, RAG is considerably cheaper than LC methods, assuming fast search is available.  Thus, understanding the tradeoffs between these methods is complex, and depends on the tradeoff between desired cost and performance for that context size.
>
> We consider our work as pushing the frontier of long-context methods.
>
> [1] Nik Pash "Why I No Longer Recommend RAG for Autonomous Coding Agents" (Substack blog post) 2025
>
> [2] Xinze Li, Yixin Cao, Yubo Ma, Aixin Sun "Long Context vs. RAG for LLMs: An Evaluation and Revisits" (arXiv:2501.01880)
>
>
> > Can the authors provide more detailed empirical or theoretical analysis on why p-RoPE pairs better with scale-invariant attention than RoPE? Frequency spectrum visualizations could help.
>
> It is understood that $p$-RoPE tends to work better than RoPE in the long-context settings by making semantic attention patterns more robust at long distances [1]. Indeed both DeepSeek and Kimi implement a form of $p$-RoPE via MLA: see in MLA that RoPE is not applied to the compressed keys and queries [2, section 2.1.1]. Further, increasing the base $\theta$, which is common practice, can be seen as approximating $p$-RoPE. We therefore believe our results on RoPE vs $p$-RoPE are simply in keeping with these more general, and well understood, trends.
>
> > While the study focuses on GPT-style models, could this method extend to encoder-decoder or encoder-only models like BERT or T5?
>
> We see no reason for our method not to extend to different transformer architectures, or indeed any NN architecture that uses softmax attention.
>
> [1]: Barbero et al., Round and round we go! what makes rotary positional encodings useful? ICLR
>
> [2]: Liu, Aixin, et al. "Deepseek-v3 technical report." arXiv.

---

> > ### Comment · Reviewer_yBmN · 2025-08-05
> >
> > Thanks for author's reply. I will retain the original score.

---

### Official Review · Reviewer_6x9F · 2025-07-01

**Clarity:** 3
**Significance:** 3
**Originality:** 4
**Rating:** 4
**Confidence:** 3

**Summary:**

This paper proposes "scale-invariant attention" to improve the ability of large language models (LLMs) to generalize from short training contexts to longer inference contexts. This method aims for attention to function stably across various scales, from local to global, and is specifically designed to satisfy two desirable properties: "scale-invariant total attention" and "scale-invariant attention sparsity". Based on a Gaussian assumption, the authors theoretically derive a method to achieve these properties by applying a simple, position-dependent affine transformation to the attention logits. In experiments, the proposed method, when combined with p-RoPE, was demonstrated to outperform prominent existing methods (such as ALiBi and LogN) in terms of validation loss. It showed particularly excellent performance in zero-shot generalization from a short context (4k) to a long context (64k). Furthermore, it maintained high performance in a long-context information retrieval task.

**Questions:**

- Regarding the IID Gaussian assumption that is central to your theory: while the QQ plot in Appendix J is interesting, do you believe the IID assumption holds even after applying position-dependent encodings like RoPE? How would you predict the effectiveness or performance of your proposed transformation to be affected if this assumption is only partially met?
- The LogN trick also aims to control the attention distribution for long contexts. Your method differs by using a position-dependent transformation and shows superior results in your experiments. Do you envision any scenarios or task types where the simpler, position-independent LogN trick might be preferable? Or does your position-dependent approach have fundamental advantages that would make it superior in almost all long-context situations?

**Ethical Concerns:**

["NO or VERY MINOR ethics concerns only"]

**Limitations:**

yes

**Quality:**

3

**Strengths And Weaknesses:**

**Strengths**
- The effectiveness of the proposed method is demonstrated through a multi-faceted and convincing set of experiments, using multiple model sizes, various training/validation context lengths, and a range of strong baseline methods for comparison. The result showing superior performance in a 16x zero-shot generalization task is particularly impressive and highlights the strength of the method.
- The proposed transformation on logits is a simple affine transformation using the logarithm of the position index, which can be easily integrated into existing Transformer architectures. This makes the research highly practical.

**Major Weaknesses**
- The theoretical derivation hinges on the assumption that attention scores follow an IID (independent and identically distributed) Gaussian distribution. The paper's justification for this assumption is insufficient. While Appendix J presents a QQ plot for a Llama-3B model, this is just a single case. It is particularly questionable whether logits remain IID with respect to position after applying positional encodings like RoPE. The main text lacks a discussion on how valid this assumption is for real-world models and what the implications would be for the method's performance if the assumption is violated.

**Minor Weaknesses**
- While Appendix H explores the lengthscale hyperparameter τ, the analysis of why τ=10 is optimal is somewhat superficial. The intuition that it represents the "size of 'chunks' we attend over" is helpful, but a more in-depth analysis of the relationship between the optimal τ, model architecture, and data characteristics would have further strengthened the paper.

---

> ### Author Rebuttal · Authors · 2025-07-30
>
> Thanks for your thoughtful and positive review!
>
> > The theoretical derivation hinges on the assumption that attention scores follow an IID (independent and identically distributed) Gaussian distribution. The paper's justification for this assumption is insufficient. While Appendix J presents a QQ plot for a Llama-3B model, this is just a single case. It is particularly questionable whether logits remain IID with respect to position after applying positional encodings like RoPE. The main text lacks a discussion on how valid this assumption is for real-world models and what the implications would be for the method's performance if the assumption is violated.
>
> After further analysis during the review period, we determined that Theorem 1 does not, in fact, require independence of Gaussians.  It simply requires marginally Gaussian random variables.  This is evident if you look at Lemmas 1+2, which do not require independent Gaussian variables, along with Appendix F, which again does not require independence.
>
> > While Appendix H explores the lengthscale hyperparameter $\tau$, the analysis of why $\tau=10$ is optimal is somewhat superficial. The intuition that it represents the "size of 'chunks' we attend over" is helpful, but a more in-depth analysis of the relationship between the optimal $\tau$, model architecture, and data characteristics would have further strengthened the paper.
>
> We agree that we established the optimal value of $\tau$ experimentally in a particular setup.
> Thus we would recommend that anyone using scale-invariant attention should check that $\tau=10$ is close-to-optimal in their setting.
> Given our intuition that $\tau$ measures the "size of 'chunks' we attend over", we would not expect the optimal $\tau$ to be too far from $10$, even in alternative architectures, but we would emphasise that this does need to be checked.
>
> #### Questions:
>
> > Regarding the IID Gaussian assumption that is central to your theory: while the QQ plot in Appendix J is interesting, do you believe the IID assumption holds even after applying position-dependent encodings like RoPE?
>
> As discussed above, IID is not necessary.  The QQ plots the QQ plots in Appendix J show attention logits in a Llama-3, to which RoPE position embeddings have been applied, and they clearly show Gaussian marginals.
>
> >The LogN trick also aims to control the attention distribution for long contexts. Your method differs by using a position-dependent transformation and shows superior results in your experiments. Do you envision any scenarios or task types where the simpler, position-independent LogN trick might be preferable? Or does your position-dependent approach have fundamental advantages that would make it superior in almost all long-context situations?
>
> We had thought that LogN might do better at a needle-in-a-haystack task, where the needle is uniformly distributed across a long context. In this setting, you would expect position-independent LogN scaling to perform better.  In practice, however, we found that scale-invariant p-RoPE performed slightly better at 16k and 64k context lengths (Table 2), but this difference was not statistically significant, so we are only able to say that performance of scale-invariant p-RoPE and LogN was indistinguishable.

---

> > ### Comment · Reviewer_6x9F · 2025-08-09
> >
> > Thank you for your response. Your clarification that the theoretical assumption only requires "marginally Gaussian" variables, not "IID", has partially addressed my concerns. The fundamental question of why the logits behave this way remains, but I understand this as an avenue for future research. Your candid responses regarding the hyperparameter τ and the comparison with LogN have also clarified my questions.

---

### Official Review · Reviewer_TuDC · 2025-07-03

**Clarity:** 3
**Significance:** 2
**Originality:** 3
**Rating:** 4
**Confidence:** 4

**Summary:**

The paper proposes to transform attention scores to achieve scale invariance. It introduces two notions of scale invariance, namely equal sum and sparsity of attention weights across token ranges. The paper derives a close-form transformation function under some assumptions that enables scale-invariant total attention, while empirically demonstrates that the same transformation also achieves scale-invariant attention sparsity. The experiments demonstrates that the training small-scale models with the proposed attention scheme achieves lower validation loss compared to the baselines, especially when the context is longer than during training. Further, the model performs better than the baselines on long-context retrieval following fine-tuning.

**Questions:**

N/A

**Ethical Concerns:**

["NO or VERY MINOR ethics concerns only"]

**Final Justification:**

The paper studies scale-invariant attention for long-context understanding. The approach is simple, theoretically grounded, and demonstrates strong results at a limited scale. While it remains unclear whether the method will yield consistent gains as model size grows, the work offers an interesting perspective on how to improve the attention mechanism to better support longer contexts.

**Limitations:**

The paper has addressed the limitations of the work.

**Quality:**

2

**Strengths And Weaknesses:**

## Strengths:
- The transformation applied on attention scores is mathematically proven to satisfy the definition of scale invariance under certain assumptions.
- The empirical results suggest the potential effectiveness of the method.

## Weaknesses:
- **Motivation:** The choice of scales (powers of 10) appears arbitrary and seems mainly for mathematical convenience. Additionally, it is unclear why a language model should satisfy the scale invariance properties. The paper asserts that scale invariance facilitates generalization on long contexts, but this claim is not substantiated by the toy experiments presented in the paper.

- **Experiments:** The paper lacks solid experiments. Validation loss is the only metric, yet there is no evidence showing its correlation with the accuracy of any downstream tasks. The NIH experiment is also not convincing, as the model is fine-tuned to overfit this very mechanical task. Ideally, one would perform scaling experiments and run a suite of LLM benchmarks to show the method works and scales well. While there may be constraints on computational resources, legitimate experiments are essential to support the claims for acceptance in a peer-reviewed conference. Nonetheless, this work could be a valuable tech report and may gain traction if the method turns out to be effective with community effort.

- **Visualization:** It helps to plot the distribution of attention weights to help visualize scale invariance.

---

> ### Author Rebuttal · Authors · 2025-07-30
>
> Thanks for your careful and positive review!
>
> >Motivation: The choice of scales (powers of 10) appears arbitrary and seems mainly for mathematical convenience.
>
> Powers of 10 are only used in the examples we use to build intuition!  They don't appear anywhere in the actual method (i.e. the scaling factors used in Eq. 16).  These scaling factors are derived from Lemma 1 and 2, which again do not use powers of 10.
>
> >Additionally, it is unclear why a language model should satisfy the scale invariance properties.
>
> This is discussed extensively in the Introduction. In short, it is clear that you don't want the LLM to focus almost exclusively on nearby tokens (1-10 tokens ago).  You also want it to look at more distant tokens.  Additionally, you don't want the LLM to focus almost exclusively on distant tokens (100-1,000,000 tokens ago), at the expense of local tokens.  But that's what standard LLMs do (see Figure 1 right hand column; orange line). You want attention to balance looking at nearby and more distant tokens. And you want the balance to be maintained as the context gets longer. That's what the scale-invariance desiderata give you. While of course the precise optimal choice of desiderata is an important topic for future work, it is clear that any attention mechanism that zero-shot generalises to long-contexts is going to have to balance attention to nearby and more distant tokens, and thus is going to have some type of scale-invariance property.
>
>
> **Experiments**
>
> We have added further experiments --- continual pretraining on Llama 2 7B at 4k context length, and evaluating the validation loss at longer context sizes (see table below) to demonstrate that scale-invariant attention extends to larger scale models.
>
> | Method                          | Val loss @ 4k | Val loss @ 16k | Val loss @ 64k |
> |--------------------------------|----------|-----------|-----------|
> | RoPE*                          | 1.968    | 7.036     | 8.815     |
> | $p$-RoPE                       | 2.029    | 3.504     | 6.800     |
> | NoPE                           | 4.152    | 6.172     | 7.730     |
> | RoPE+NTK*                      | 1.988    | 3.090     | 7.523     |
> | YaRN                           | 1.957    | 7.004     | 8.763     |
> | LogN+RoPE                      | 1.966    | 6.932     | 8.726     |
> | LogN+$p$-RoPE                  | 2.049    | 2.984     | 6.224     |
> | LogN+NTK                       | 1.966    | 3.063     | 7.413     |
> | ALiBi                          | 2.750    | 2.745     | 2.744     |
> | Scale-invariant $p$-RoPE (ours)| 2.163    | 2.193     | 2.252     |
>
> Specifically, we continually pretrained on Llama 2 at 4k context length after replacing the default attention mechanism with various other methods, and we looked at validation loss at out of distribution lengths (4k/16k/64k). Changing the attention mechanism degraded the loss at the training context length, but scale-invariant $p$-RoPE far exceeded the other methods. Note that we chose Llama 2 for this continual pretraining experiment because it is one of the few models that have not already mid-trained at a longer context length.
>
>
> With this addition, our experiments used around 500 H100 hour equivalents. This is a typical scale for an academic paper published at an academic conference, and indeed is similar to the compute used in previous academic papers presented at academic conferences presenting new long-context attention mechanisms (e.g. [1,2,3]).  It is not at all clear we should be demanding authors have foundation-model-lab compute budgets, especially given that these labs are increasingly refusing to publish important innovations.
>
>
>
> > Visualization: It helps to plot the distribution of attention weights to help visualize scale invariance.
>
> We agree! We will add this for the camera ready.
>
> Refs:
>
> [1]: Press et al., Train short, test long: Attention with linear biases enables input length extrapolation
>
> [2]: Nakanishi, Scalable-softmax is superior for attention
>
> [3]: Li et al., Information entropy invariance: Enhancing length extrapolation in attention mechanisms

---

### Comment · Area_Chair_t3ZW · 2025-08-05

Dear Reviewers, this is a gentle reminder to read the authors' rebuttal and the other reviews carefully. You are encouraged to post your initial response and engage in the discussion with the authors. This author-reviewer discussion period ends on August 6th AoE. Thank you.

---

> ### Comment · Area_Chair_t3ZW · 2025-08-05
>
> Dear Reviewers, the author-reviewer discussion period has been extended to August 8th AoE. Please use this extra time to read the author's rebuttal and engage in discussion with the authors. Thank you.

---

### Decision · Program_Chairs · 2025-09-17

**Decision:**

Accept (poster)

**Comment:**

This paper proposes scale-invariant attention, a mechanism to improve LLMs' ability to generalize from short training contexts to much longer inference contexts. The authors rigorously define two key properties: scale-invariant total attention and scale-invariant attention sparsity—and mathematically derive a simple, position-dependent logit transformation to satisfy them. The method is empirically validated through a series of experiments, showing strong zero-shot generalization and outperforming baselines like YaRN and LogN on out-of-distribution context lengths.

The paper stands out for its strong theoretical foundation, moving beyond heuristic methods by deriving its approach from first principles. The proposed logit transformation is both theoretically sound and highly practical, as it is simple to implement and integrates easily with existing Transformer architectures. The empirical results, particularly the impressive zero-shot generalization for long-context setting, provide compelling evidence of the method's potential for real-world impact.

The primary initial weakness was the limited scale of the experiments, which was a concern for Reviewer TuDC, Reviewer yBmN, and Reviewer tNeo. This raised questions about the method's applicability to much larger, commercially deployed models. Another point of theoretical weakness is the reliance on the Gaussian assumption for the attention logits, a property that is empirically observed but not yet fully understood, as noted by Reviewer 6x9F and Reviewer tNeo.

I recommend acceptance because the paper presents a significant and well-supported contribution to the field. The authors successfully addressed the most critical concern regarding scalability. The zero-shot generalization performance is compelling and demonstrates the method's potential to enable cost-efficient LLM deployment.

All reviewers initially raised concerns, primarily focused on the limited experimental scale and the robustness of the theoretical assumptions. In their rebuttal, the authors provided new experiments on a 7B Llama 2 model, which effectively addressed the scalability concerns raised by Reviewer TuDC, Reviewer yBmN, and Reviewer tNeo. The authors also clarified the theoretical assumptions to Reviewer 6x9F and Reviewer tNeo, explaining that their theory only requires "marginally Gaussian" variables. Lastly, the authors addressed the lack of comparisons with Infini-Attention and Ring Attention raised by Reviewer tNeo, providing new results against the former and a clear explanation for why the latter is a different class of method.